# Resting-State Functional MRI in Dyslexia: A Systematic Review

**DOI:** 10.3390/biomedicines13051210

**Published:** 2025-05-16

**Authors:** Bruce Martins, Isabel A. B. Verrone, Mariana M. I. Sakamoto, Mariana Y. Baba, Melissa E. Yvata, Katerina Lukasova, Mariana P. Nucci

**Affiliations:** 1LIM44—Hospital das Clínicas da Faculdade Medicina da Universidade de São Paulo, São Paulo 05403-000, Brazil; bruce.martins@hc.fm.usp.br (B.M.); bender.isabel@ufabc.edu.br (I.A.B.V.); mari.ikoma@gmail.com (M.M.I.S.); mariana.yumi@hc.fm.usp.br (M.Y.B.); melissa.yvata@fm.usp.br (M.E.Y.); 2Centro de Matemática, Computação e Cognição (CMCC), Universidade Federal do ABC, Santo André 09210-580, Brazil; katerina.lukasova@ufabc.edu.br

**Keywords:** dyslexia, resting state, rs-fMRI, neurodevelopment, functional connectivity

## Abstract

**Background/Objectives:** The present review addresses and systematically analyses the most frequently reported neuropsychological and functional connectivity (FC) alterations in individuals with dyslexia compared to controls. By synthesizing extant evidence, this work aims to clarify dyslexic connectivity profiles and provide a foundation for future research and clinical translation. **Methods:** This systematic review analyzed publications from the last 10 years in two scientific databases, focusing on individuals with dyslexia, without previous injuries, who underwent resting-state functional magnetic resonance imaging (rs-fMRI) assessments, comparing them with typical readers. **Results:** This review revealed that most dyslexia studies on brain FC using rs-fMRI focused on children (92%), underscoring a gap in research on adults and limiting our understanding of brain maturation processes and neuroplasticity across the lifespan. FC alterations primarily involved ipsilateral connections (60%), with reduced connectivity mainly in the left hemisphere (40%), particularly in posterior regions, aligning with the neurobiological hypothesis of phonological and visual–phonological dysfunctions in dyslexia. Conversely, increased connectivity in the right hemisphere (20%) may indicate the engagement of an alternative network and highlight the complexity of neural adaptations in dyslexia. **Conclusions:** The findings highlight a significant gap in the study of adult dyslexia and suggest that FC alterations predominantly affect the left hemisphere, with possible compensatory mechanisms in the right hemisphere. Reading fluency improvements in dyslexia may be linked to connectivity changes across multiple brain networks rather than the classical reading circuitry alone. Increased and decreased connectivity in various regions related to executive function, language, and salience processing indicate that broader cognitive mechanisms play a key role in reading performance.

## 1. Introduction

Reading fluency, the ability to read accurately and effortlessly, is a critical skill for optimal individual development in our modern society [1]. This skill is essential for reading success, as it allows readers to devote their cognitive resources from word decoding to comprehension [2]. Systematic disturbances in this process, such as in developmental dyslexia, are worthy of significant scientific attention, particularly given their prevalence. Developmental dyslexia is defined by both the Diagnostic and Statistical Manual of Mental Disorders 5th Edition (DSM-5) and International Classification of Diseases 11th Revision (ICD-11) as a specific learning disorder marked by persistent difficulties in reading fluency, accuracy, decoding, and comprehension, despite adequate intelligence, sensory function, and educational exposure. According to the DSM-5 [3], it is classified under “Specific Learning Disorder with impairment in reading”, while the ICD-11 [4] characterizes it as a developmental disorder of reading skills. Both frameworks emphasize that symptoms should emerge early in schooling, cause functional impairment, and not be attributable to other neurological or psychiatric conditions. This standardized approach ensures consistency in the clinical and neuroimaging characterization of dyslexia, which affects reading specifically without impairing broader cognitive domains like attention, memory, or intelligence. [3,4]. It is estimated to be prevalent in approximately 7% of the world population [5], with higher prevalence in boys compared to girls [6]. Yet, understanding and addressing dyslexia remains a crucial area for ongoing research and development of intervention strategies [7].

The application of various magnetic resonance imaging (MRI) techniques, including structural MRI, diffusion MRI, and functional MRI (fMRI) [8], has emerged as one of the main approaches over the past decades to advance our understanding of the underlying mechanisms and root causes of this disorder, and provide complementary insights into brain characteristics of the condition [9]. Out of these, fMRI plays a central role in determining which cortical networks and regions of interest (ROIs) are recruited in individuals with dyslexia when engaged in tasks or resting [8,9]. Rs-fMRI, especially, makes it possible to assess FC, since it explores spontaneous fluctuations in brain activity that reflect the functional organization of cortical networks and their interactions [10].

As commonly observed in the literature, the vast majority of functional neuroimaging studies in dyslexia focus on children, while reports involving adults with dyslexia remain scarce. This pattern was also noted in our previous systematic review on brain structural alterations across the lifespan in individuals with dyslexia [11]. In pediatric populations, findings related to brain functional connectivity (FC) are often inconsistent, largely due to the use of diverse methodological approaches—most involving task-based fMRI—as well as variations in the selection of neural networks and regions of interest (ROIs) analyzed.

Nonetheless, multiple studies have reported that, compared to age-matched controls, children with dyslexia exhibit altered FC in reading-related networks, including the sensorimotor dorsal, frontoparietal, visual, and auditory networks [12]. These alterations typically involve increased connectivity in regions associated with cognitive control and decreased connectivity in language and subcortical regions. In adults, although the literature is limited, the few available studies [13] show patterns that are partially consistent with findings in children. These include both disruptions in networks critical for reading and the emergence of compensatory mechanisms, particularly in executive control systems such as the frontoparietal and salience networks.

Despite the growing body of research on dyslexia, a gap persists in the systematic synthesis of rs-fMRI findings specific to this population. To date, no comprehensive review has consolidated the heterogeneous methodologies and variable results reported across studies. The present review addresses this gap by systematically analyzing the most frequently reported neuropsychological and FC alterations in individuals with dyslexia compared to controls. By synthesizing extant evidence, this work aims to clarify dyslexic connectivity profiles and provide a foundation for future research and clinical translation.

## 2. Materials and Methods

### 2.1. Information Sources and Search Strategy

A systematic literature search was conducted using two electronic databases: PubMed and Scopus. The search encompassed studies published between 2015 and 2025, with the final search executed on 16 January 2025. The selection and reporting of studies followed the Preferred Reporting Items for Systematic Reviews and Meta-Analyses (PRISMA) guidelines (https://www.prisma-statement.org/; accessed 16 January 2025) [14]. This study was registered on Prospero 2025 (CRD420251024961).

Search strategies were developed using predefined Boolean operators and keywords relevant to dyslexia and functional connectivity. The full search strings used were as follows:

PubMed: (dyslexia [Title/Abstract] OR “Reading disorder” [Title/Abstract] OR “Reading disorders” [Title/Abstract] OR “Reading disability” [Title/Abstract] OR “Reading disabilities” [Title/Abstract] OR “Developmental reading disability” [Title/Abstract] OR “Developmental reading disabilities” [Title/Abstract] OR “Developmental reading disorder” [Title/Abstract] OR “Developmental reading disorders” [Title/Abstract] OR “developmental dyslexia” [Title/Abstract] OR “reading deficit” [Title/Abstract] OR “reading impairment” [Title/Abstract] OR “developmental language disorder” [Title/Abstract] OR “dyslexia reading” [Title/Abstract]) AND (“resting state” [Title/Abstract] OR “resting-state” [Title/Abstract] OR rs-fmri [Title/Abstract] OR “functional connectivity” [Title/Abstract] OR “resting state fMRI” [Title/Abstract] OR “functional brain connectivity” [Title/Abstract] OR “resting-state fMRI” [Title/Abstract] OR “magnetic resonance imaging” [Title/Abstract]).

Scopus: (TITLE-ABS-KEY (dyslexia OR “Reading disorder” OR “Reading disorders” OR “Reading disability” OR “Reading disabilities” OR “Developmental reading disability” OR “Developmental reading disabilities” OR “Developmental reading disorder” OR “Developmental reading disorders” OR “developmental dyslexia” OR “reading deficit” OR “reading impairment” OR “developmental language disorder” OR “dyslexia reading”) AND TITLE-ABS-KEY (“resting state” OR “resting-state” OR rs-fmri OR “functional connectivity” OR “resting state fMRI” OR “functional brain connectivity” OR “resting-state fMRI” OR “magnetic resonance imaging”)) AND PUBYEAR > 2014 AND PUBYEAR < 2026 AND (EXCLUDE (EXACTKEYWORD, “EEG”) OR EXCLUDE (EXACTKEYWORD, “Case Report”) OR EXCLUDE (EXACTKEYWORD, “Child”) OR EXCLUDE (EXACTKEYWORD, “Cerebrovascular Accident”) OR EXCLUDE (EXACTKEYWORD, “Glioma”) OR EXCLUDE (EXACTKEYWORD, “Brain Neoplasms”) OR EXCLUDE (EXACTKEYWORD, “Schizophrenia”) OR EXCLUDE (EXACTKEYWORD, “Disease Association”) OR EXCLUDE (EXACTKEYWORD, “Primary Progressive Aphasia”) OR EXCLUDE (EXACTKEYWORD, “Seizure”) OR LIMIT-TO (EXACTKEYWORD, “Controlled Study”) OR EXCLUDE (EXACTKEYWORD, “Alzheimer Disease”) OR EXCLUDE (EXACTKEYWORD, “Nonhuman”) OR EXCLUDE (EXACTKEYWORD, “Alzheimer’s Disease”) OR EXCLUDE (EXACTKEYWORD, “Dementia”) OR EXCLUDE (EXACTKEYWORD, “Parkinson Disease”)) AND (LIMIT-TO (SUBJAREA, “MEDI”) OR LIMIT-TO (SUBJAREA, “NEUR”) OR LIMIT-TO (SUBJAREA, “PSYC”)) AND (LIMIT-TO (DOCTYPE, “ar”)) AND (LIMIT-TO (LANGUAGE, “English”)) AND (LIMIT-TO (SRCTYPE, “j”)) AND (EXCLUDE (EXACTSRCTITLE, “Journal Of Alzheimer S Disease”) OR EXCLUDE (EXACTSRCTITLE, “Alzheimer S And Dementia”) OR EXCLUDE (EXACTSRCTITLE, “Frontiers In Oncology”) OR EXCLUDE (EXACTSRCTITLE, “Aphasiology”) OR EXCLUDE (EXACTSRCTITLE, “Epilepsia”) OR EXCLUDE (EXACTSRCTITLE, “Epilepsia Open”) OR EXCLUDE (EXACTSRCTITLE, “Current Alzheimer Research”) OR EXCLUDE (EXACTSRCTITLE, “Alzheimer’s And Dementia Diagnosis Assessment And Disease Monitoring”) OR EXCLUDE (EXACTSRCTITLE, “Parkinsonism And Related Disorders”) OR EXCLUDE (EXACTSRCTITLE, “Epilepsy Research”) OR EXCLUDE (EXACTSRCTITLE, “Epilepsy Currents”) OR EXCLUDE (EXACTSRCTITLE, “BMC Musculoskeletal Disorders”) OR EXCLUDE (EXACTSRCTITLE, “Autism Research”) OR EXCLUDE (EXACTSRCTITLE, “Stroke”) OR EXCLUDE (EXACTSRCTITLE, “Future Oncology”) OR EXCLUDE (EXACTSRCTITLE, “Multiple Sclerosis And Related Disorders”) OR EXCLUDE (EXACTSRCTITLE, “Cancers”) OR EXCLUDE (EXACTSRCTITLE, “Brain Injury”) OR EXCLUDE (EXACTSRCTITLE, “Amyotrophic Lateral Sclerosis And Frontotemporal Degeneration”) OR EXCLUDE (EXACTSRCTITLE, “Dementia And Geriatric Cognitive Disorders Extra”) OR EXCLUDE (EXACTSRCTITLE, “Cancer”) OR EXCLUDE (EXACTSRCTITLE, “Alzheimer’s Dementia The Journal Of The Alzheimer’s Association”) OR EXCLUDE (EXACTSRCTITLE, “Alzheimer’s And Dementia Translational Research And Clinical Interventions”)).

### 2.2. Selection Criteria

The review included original research articles published in peer-reviewed journals, written in English, and published within the last 10 years. Study selection was guided by the PICO (population, intervention, comparison, outcome) framework as the primary inclusion criterion. The population (P) encompassed individuals of any age diagnosed with dyslexia or reading disorders, excluding those with a history of brain injury or other neurological conditions. Intervention (I) referred to studies employing rs-fMRI to assess functional brain alterations associated with dyslexia. Comparison (C) required the inclusion of a control group composed of individuals with typical reading abilities for direct comparison with the dyslexic group. Outcome (O) involved the reporting of results either through group comparisons or through correlations between FC and demographic or neuropsychological variables.

### 2.3. Exclusion Criteria

The exclusion criteria for this review comprised articles classified as narrative reviews, systematic reviews, descriptive reviews, meta-analyses, case reports, expert opinions, and book chapters, as the focus was exclusively on original research. Studies not published in English were excluded to ensure consistency in data interpretation.

Research involving participants with reading disorders resulting from neurological or psychiatric comorbidities—such as stroke, traumatic brain injury, epilepsy, Alzheimer’s disease, dementia, or Parkinson’s disease—was excluded. Similarly, studies that included participants with reading disorders associated with other conditions, such as autism, aphasia, and mood disorders, or that focused on familial history or genetic alterations related to dyslexia (e.g., in neonates), were not considered. In addition, studies that did not include a comparison between individuals with dyslexia and typically developing readers were excluded.

Concerning the intervention, studies using imaging methodologies other than rs-fMRI—such as EEG, MEG, or fNIRS—were excluded. Studies employing analytical approaches outside the scope of conventional FC analysis, including those based solely on machine learning techniques, were also excluded. Furthermore, studies that analyzed resting-state conditions within task-based fMRI designs, as well as those focused on structural connectivity rather than FC, were not included.

### 2.4. Selection Process and Data Compilation

The selection process for the articles included in this systematic review was conducted by four authors (B.M., M.M.I.S., I.B. and M.Y.B.) using the Rayyan software (https://www.rayyan.ai/) [15]. Initially, duplicated references were automatically identified and removed. Subsequently, the titles and abstracts of the remaining studies were independently screened by the authors M.M.I.S., I.B., M.Y.B., and M.E.Y., working in randomly assigned pairs, in a blinded fashion, and according to predefined inclusion and exclusion criteria. In cases of disagreement between reviewers on the inclusion of any article, a third senior author (M.P.N. or K.L.) was consulted to resolve the conflict and reach a consensus. Articles deemed potentially eligible were retrieved in full text and reassessed by the entire team for final eligibility. The final selection was based on group consensus among all participating authors (B.M., M.M.I.S, I.B., M.Y.B., M.E.Y., K.L. and M.P.N.).

As part of the screening process, keyword-based filtering was applied to identify studies that met the eligibility criteria. Keywords were used to include relevant studies and exclude those that did not align with the review’s objectives. Additionally, studies were manually labeled and excluded based on methodological or thematic aspects that did not fit the scope of the review. This process ensured that only studies investigating resting-state functional connectivity (rs-FC) in dyslexia were included. This rigorous selection process ensured that only relevant studies investigating rs-FC in individuals with dyslexia were included in the final analysis.

The authors M.M.I.S., I.B., and M.E.Y. were responsible for extracting the demographic, clinical, and neuropsychological characteristics of volunteers and dyslexia patients, with verification by B.M. They also compiled the characteristics of functional brain imaging (a parameter of acquisition and data analysis) and their outcomes, checked by M.Y.B. All data were double checked by the senior authors K.L. and M.P.N.

The conceptualization of this systematic review was undertaken by B.M., K.L., and M.P.N. The authors M.M.I.S., I.B., B.M., M.Y.B., and M.E.Y. were responsible for conducting the review and analysis, while the preparation of the original draft was carried out by M.M.I.S. and I.B. The review and editing of the manuscript were undertaken by B.M., M.Y.B., K.L., and M.P.N. The overall supervision and management of this project were the responsibility of M.P.N. All of the authors contributed to the writing of the complete manuscript.

### 2.5. Data Collection

Two main topics were used to organize the data from the papers under review, and the corresponding information was presented in tables highlighting the following features. The demographic and neuropsychological domain, summarized in the first table, included the language and nationality assessed in each study, an essential aspect given that dyslexia manifestations may vary across languages, as well as detailed characteristics of the study population, such as sample size, age, sex, and years of education for both dyslexic and control participants. It also describes the neuropsychological tests used to assess intelligence, reading ability, and other cognitive domains to rule out comorbidities such as attention deficits, memory impairments, and motor or oculomotor coordination issues. The neuroimaging methodology domain, presented in the second table, detailed how rs-fMRI data were acquired and preprocessed, as well as the FC measures applied. It also reported the main findings regarding group differences and correlations with clinical and neuropsychological data, particularly those related to reading processes. This structured approach provides a comprehensive descriptive overview of the neural mechanisms underlying dyslexia, enabling better comparability and interpretation across studies.

### 2.6. Risk-of-Bias Assessment

The article selection process was conducted in pairs, with a third independent author responsible for determining their inclusion. The data presented in the tables were organized into the predefined groups by the authors, and their accuracy was validated by an independent group. The final inclusion of studies in the systematic review was decided through consensus among all reviewers.

### 2.7. Methodological Quality Assessment

The methodological quality of the included studies was assessed using the Newcastle–Ottawa Scale (NOS) [16], a widely adopted tool for evaluating the risk of bias in observational studies. The NOS evaluates three main domains: the selection of study groups, the comparability of groups, and the ascertainment of exposure (or outcome).

The selection domain considers the adequacy of case definitions, sample representativeness, the selection of control groups, and the clear exclusion of the target condition in controls. Comparability examines whether the study controlled for major confounding factors relevant to the research question, either through design or statistical analysis. The exposure section assesses the reliability of exposure measurements, the consistency of methods between cases and controls, and non-response rates.

Each study can receive a maximum score of 9 points, with high methodological quality defined as scores between 7 and 9, moderate quality between 4 and 6, and low quality between 0 and 3 points [17].

The NOS was independently applied by three reviewers (BM, IABV, and MYB). To evaluate the consistency between raters, the Cohen’s Kappa coefficient was calculated, ranging from 0 (no agreement) to 1 (perfect agreement) [18]. This was computed using R Statistical Software (v2021; https://www.r-project.org; accessed 1 May 2015) and the statistical package “irr” (v0.84.1; R Core Team 2019; https://www.r-project.org, accessed 1 May 2025) [19].

### 2.8. Data Analysis

The data presented in each table were analyzed using percentage values and range distributions to emphasize key characteristics, specific patterns, and notable exceptions according to their respective applications. Additionally, the results were categorized to highlight the most salient aspects within each context, providing an integrated overview of the systematic review through both quantitative and qualitative analyses.

## 3. Results

### 3.1. Study Selection

A total of 1025 references were imported from the selected databases. The majority of these references, 822 studies, were retrieved from Scopus, while 203 references were imported from PubMed, contributing to the final dataset used for screening and selection.

From the 822 articles identified in Scopus, 817 were excluded during screening. The primary reasons for exclusion were irrelevance to developmental dyslexia (285 studies, of which 212 did not address dyslexia and 73 focused on acquired dyslexia secondary to brain injury or neurological conditions); duplication or non-original research (203 studies, comprising 186 duplicates, 16 reviews/meta-analyses, and 1 opinion piece); ineligible study designs or populations (106 studies, including 98 case reports, 6 studies without a control group, and 2 neonatal studies); and incompatible neuroimaging modalities (211 studies, of which 84 used non-rs-fMRI techniques [e.g., EEG, MEG, or fNIRS], 61 employed task-based fMRI, 42 analyzed only structural connectivity, and 24 lacked FC analyses). An additional 12 studies were excluded for using machine learning-only approaches. After that, two articles were removed in the eligibility process because they did not report application of any neuropsychological tests; after exclusions, three articles met the inclusion criteria. Similarly, of the 203 articles retrieved from PubMed, 179 were excluded. These exclusions mirrored the Scopus criteria, with the largest categories being incompatible neuroimaging modalities (91 studies: 29 using non-rs-fMRI, 53 task-based fMRI, 10 structural connectivity-only, and 9 lacking FC analyses) and irrelevance to developmental dyslexia (32 studies: 29 not addressing dyslexia and 3 involving acquired dyslexia). Other exclusions included non-original research (24 reviews, 1 opinion article, 1 book chapter), ineligible designs (10 case reports, 2 studies without controls), and 8 machine learning-only studies. After that, 2 articles were removed in the eligibility process because they did not compare the control group with the dyslexics; after exclusions, 22 articles met the inclusion criteria. After completing the selection process and applying all exclusion criteria across both databases, only 25 full original articles were included in this systematic review. With all the exclusions from both databases, there were only 25 articles left [20,21,22,23,24,25,26,27,28,29,30,31,32,33,34,35,36,37,38,39,40,41,42,43,44], as illustrated in the PRISMA flowchart (Figure 1).

### 3.2. Methodological Quality Assessment Results

Three independent reviewers assessed the methodological quality of the included studies using the Newcastle–Ottawa Scale (NOS). Inter-rater agreement varied among the pairs: Rater 1 and Rater 2 agreed on 48% of the ratings (Cohen’s kappa = 0.329, *p* = 0.001), indicating fair agreement; Rater 1 and Rater 3 agreed on 36% (kappa = 0.206, *p* = 0.024); and Rater 2 and Rater 3 agreed on 64% (kappa = 0.545, *p* < 0.001), reflecting moderate agreement. To determine the final methodological quality rating for each study, a majority rule was applied: at least two raters had to agree on the rating for it to be accepted. Based on this process, of the 25 studies evaluated, 4 were classified as high methodological quality, 19 as moderate, and 2 as low quality according to the NOS criteria. This approach ensured that the final ratings reflected a consensus among the reviewers, thereby enhancing the reliability of the quality assessment (Figure 2).

### 3.3. Study Characteristics

#### 3.3.1. Demographic Data

In the context of this systematic review, Table 1 serves as a key resource for understanding the demographic and neuropsychological characteristics of individuals with developmental dyslexia studied in rs-FC analysis. Given the variability of dyslexia across age groups and linguistic backgrounds, this table organizes the selected studies according to key demographic variables, allowing for a more nuanced understanding of the neurobiological aspects of the disorder. By categorizing participants into children and adults, the table enables the analysis of developmental differences and how the manifestations of dyslexia may change over time, highlighting potential age-related neural patterns and adaptations.

Moreover, including neuropsychological assessments highlights the range of cognitive functions examined in these studies, shedding light on how reading proficiency, phonological processing, and attention mechanisms may be linked to underlying connectivity differences.

Table 1 provides a list of the countries and languages represented in the dataset. English is the most frequently studied language, appearing in 14 of the selected studies (56.0%) [22,26,28,29,30,31,32,35,37,38,39,41,42,43]. German appears twice (8.0%) [33,44], while French [24], Turkish [21], Brazilian Portuguese [36], and Catalan [27] are represented only once (4.0% each). Additionally, Japanese is noted with both its syllabic Hiragana and logographic Kanji scripts (4.0%) [34]. If split into opaque and transparent languages (according to the phoneme–grapheme correspondence) [45], German, Turkish, Brazilian Portuguese, Catalan, and Japanese Hiragana (syllabic scripts are highly phonetic and transparent) are part of the transparent group, comprising 36% of the sample, and the opaque group consisting of English, Chinese, French, and Japanese Kanji are 64% of the sample. The literature shows that dyslexia diagnosis is easier in transparent languages because consistent letter–sound mapping allows for easier decoding of words, and difficulties related to dyslexia are more evident and not mistaken as reading errors or poor comprehension of irregularities in the language [46].

The United States (USA) appears most frequently, with 13 mentions (52.0%) [22,26,28,29,31,32,35,37,38,39,41,42], followed by China with 4 occurrences (16.0%) [20,23,25,40], while Canada [30,43] and Germany [33,44] each appear twice (8.0% each), and Turkey [21], Belgium [24], Brazil [36], Spain [27], and Japan are mentioned only once [34] (4.0% each), as shown in Table 1. In terms of language distribution, there is a strong research focus on English and studies conducted in the USA, with other languages and regions remaining comparatively underrepresented.

It is important to note that many studies use different nomenclatures when they are talking about dyslexia, such as “reading disorders”, “learning impairment”, “reading impairment” and other variations of the sort. In order to simplify understanding, we have placed all these terms under the umbrella of dyslexia. All studies contained a control group and a dyslexic group with some variations in regard to additional groups; we used abbreviations to simplify the table. There are a plethora of terms and abbreviations used to describe dyslexia, but all of the studies finalize the diagnostics as dyslexia once you evaluate the neuropsychological tests. For comprehension and clarity purposes, we chose the umbrella term dyslexia and the abbreviation DYS (dyslexia); for specific subgroups in specific studies, we have DYSe (dyslexics that receive special education), MD (mood disorders), and BD (behavior disorders). Specific subgroups represent a minority of the studies, accounting for only 8.0% (2 out of 25) [21,37], as shown in Table 1.

The number of participants in each study is often split between control and dyslexic groups, with variations in sample sizes across studies. The mean number of subjects is 26 controls and 25 DYS, with the biggest study with 65 controls and 55 DYS and the smallest with 9 controls and 10 DYS (Table 1). Most of the selected studies included both male and female subjects, with only one study exclusively involving male subjects. Notably, only three studies reported a majority of female participants with dyslexia.

Regarding the age of participants in the selected studies, the values ranged from children to young adults. To facilitate the analysis of the neuropsychological assessment tools employed, the content following Table 1 was organized into two groups: first, the selected studies involving children, followed by those involving adults. This structure allows for a clearer evaluation of the tools used, considering the necessary adaptations based on age. The majority of studies (92% (23 out of 25) focused on children, with ages ranging from pre-readers at 5 years old to pre-teens to 12 years old [20,21,22,23,24,25,26,27,28,29,30,31,32,33,34,35,36,37,38,39,40,41,42]. As commonly observed in the dyslexia literature, studies involving adults are less frequent. Among the selected studies, only two included adult participants [43,44], one reported on teens aged 17–18 years, and the other on young adults (21–22 years old). This inclusion reflects an emergence not only in understanding the patterns of functional brain connectivity associated with dyslexia but also in exploring age-related neuroplasticity across different developmental stages.

Considering the reading age, most of the selected studies did not report this information (Table 1). Only a few studies [20,21,25,27,33] classified participants according to literacy stages; for example, one longitudinal study divided its sample into two groups: “before literacy introduction” (BL) and “after literacy introduction” (AL). Similarly, information on years of education was absent in 92.0% (23/25) of the studies [20,21,22,23,24,26,27,28,29,30,31,32,34,35,36,37,38,39,40,41,42,43,44], which is expected given the predominant focus on children. However, some studies did specify educational background, reporting, for instance, prior exposure to dyslexia-focused instruction lasting 12–24 months or categorizing participants by school level, such as kindergarten, second grade, or 4th–6th grade.

#### 3.3.2. Neuropsychological Tests for Dyslexia Evaluation

According to the diagnostic criteria, individuals with dyslexia must exhibit impaired reading performance while other cognitive domains remain intact. Furthermore, these deficits should not be attributable to other factors such as sensory impairments or inadequate education [22]. To confirm this, non-verbal intelligence tests are often employed to verify that the individual’s intellectual abilities are comparable to those of people without dyslexia. Differentiation between dyslexic and typical readers typically involves reading and/or phonological tests. Dyslexia is also highly comorbid with conditions such as ADHD, dyscalculia, and dyspraxia. As a result, assessments for attention, mathematical abilities, and motor skills are commonly included in the diagnostic process. It is important to note that most diagnostic tools are language-specific and require adaptations and validation tailored to the linguistic and cultural context of each country. Standardized assessments are frequently modified from their original versions to ensure they are appropriate for specific populations.

Intelligence tests were administered across the studies, employing diverse methodologies. The most frequently used test was the WISC (32.0%, 8/25) [20,21,24,26,27,32,34,39], in various editions (WISC-IV and WISC-III). The TONI was the second most common (28.0%, 7/25) [22,28,29,31,38,41,42], with TONI-III and IV being the versions applied. Other frequently used assessments included the WASI (20.0%, 5/25) [26,30,36,39,43]; verbal, nonverbal, and full-scale versions) and Raven’s Progressive Matrices (nonverbal; 16.0%, 4/25) [23,25,34,40]. Less common measures were the PPVT (11.1%, 3/25) [28,31,38] and the WPPSI (4.0%, 1/25) [33], with one study not employing an IQ test [37]. The predominance of nonverbal intelligence assessments (76.0%, 19/25) [22,23,25,26,28,29,30,31,33,34,35,38,40,41,42,43,44] suggests a methodological emphasis on evaluating cognitive abilities independent of language proficiency, since the verbal component of intelligence may be influenced by dyslexia’s core deficits.

Reading accuracy was assessed using a range of standardized measures. Word reading was evaluated in the majority of studies (92.0% 23/25) [20,21,22,24,25,26,27,28,29,30,31,32,33,34,35,36,37,38,39,41,42,43,44]; the most frequently used tests were the Test of Word Reading Efficiency (TOWRE) [3,7,9,10,11,12,13,16,22,23,24] and the Woodcock–Johnson (WJ) [22,26,28,29,31,32,35,37,38,39,44], each appearing in 44.0% (11/25) of the studies. The most common subtests of these two were the WJ Letter–Word Identification subtest [6,22,26,28,29,31,35,37,38,39,44] and the TOWRE Sight Word Efficiency subtest (8/11) [6,22,26,28,29,30,31,35,43]. The Chinese Character Recognition Test Battery (CRTB) was used in 8.0% (2/25) of studies [20,25], underscoring language-specific adaptations in word recognition assessment. Less frequently used tests included the Test of Adolescent and Adult Language (TALE-C) [27], the Test of Silent Reading Efficiency (TOSREC) [28], the Salzburger Lese- und Rechtschreibtest (SLRT-II) [33], the Word Reading Task (LPI) [36], the Word Identification Task [43], and the Standard Comprehension Test [34], each appearing in 4.0% (1/25) of the studies. Pseudoword reading accuracy was evaluated in a large portion of the studies (72.0%, 18/25) [22,24,26,27,28,29,30,31,32,35,36,37,38,39,41,42,43,44], with various standardized tests applied. The TOWRE was the most frequently used measure, appearing in 44.0% (11/25) [22,26,28,29,30,31,32,39,41,42,43], followed by the Word Attack subtest of WJ-III, which appeared in 32.0% of the studies (8/25) [26,28,29,35,37,38,39,43]. PROLEC-R Pseudowords [27], LPI Pseudoword Reading Task [36], and the Analytic Battery Nonwords [24] each appear in 1 paper (4.0%). The majority of studies reported significant group differences in word reading (78.26%, 18/23 total with word reading tests) and pseudoword (83.33%, 15/18 total with pseudoword reading tests) reading performance between dyslexic and control participants. These results are expected given the population being studied, and also are suggestive of the correct diagnosis of the dyslexic group.

More than half of the studies (48.0%, 12/25) report evaluating text reading accuracy [23,25,26,27,28,29,30,31,32,38,39,42,44]. Among those, the most frequently used measures were Woodcock–Johnson subtests (passage comprehension and reading fluency), employed in 20.0% (5/25) of the studies [28,29,30,31,39,44]. The second most common assessment was the Gray Oral Reading Test (GORT) (20.0%, 5/25) [26,29,32,39,42]; in some cases, both WJ and GORT were administered together [29,39]. Other tests included the Test of Silent Reading Efficiency (TOSREC) (8.0%, 2/25) [29,38] and the TALE-C text subtest (4.0%, 1/25) [27]. Additionally, unspecified reading fluency tests (8.0%, 2/25) and a paragraph reading task (4.0%, 1/25) were reported. The grand majority of papers who evaluated text reading (11/14, 84.6%) [23,25,27,28,29,30,31,32,39,42] reported significant group differences between dyslexics and controls in text reading assessments. These assessments focused on different aspects of reading, including fluency and comprehension, providing insights into how dyslexia affects text processing beyond single-word recognition and also ensuring a more naturalistic investigation of reading difficulties.

Tests aimed to measure phonemic awareness and sound manipulation were also applied. They are important for the development of reading skills as they enable individuals to identify and manipulate phonemes. Phonological processing was assessed 44.0% of the time (11/25) [22,23,28,29,31,34,37,38,40,41,44], with the Elision subtest from the Comprehensive Test of Phonological Processing (CTOPP) appearing in 24.0% (6/25) [22,28,29,31,37,38] of the studies. The Rapid Automatized Naming test (RAN) was included in a number of studies (44.0%, 11/25) [20,27,28,30,31,33,34,39,40,41,44], with a variation of the full application (Digits, Letters, Colors, and Shapes) or just one part of the test. Most applications showed significant differences between control and dyslexics performance in the task (63.64%, 7/11) [20,28,30,31,34,40,41,44].

Other domains were assessed, such as visual attention, motor skills, and memory, mostly as secondary measures to rule out other comorbidities. Language and attention assessments were included in a smaller portion of the studies. Language evaluations were only 4% of studies (1/25), featuring the Vocabulary Test (CRM) [23]. Similarly, attention assessments were conducted in 32% of the studies (8/25) [20,28,29,31,35,38,41,42]; the most common measure was Conners Comprehensive Behavior Rating Scales (CBRS) and its subtests (24% 6/25) [20,28,29,38,41,42]. After that, there were four articles (16%) that used the Test of Everyday Attention (TEA-Ch) [28,31,35,41], and there was also a individual case of Continuous Performance Test (CPT) [28]. Inhibition tasks, such as the Stroop test, were employed in 16.0% (4/25) [22,28,31,38,41] of studies, while parent/teacher-reported EF (BRIEF) and performance-based flexibility tests (D-KEFS, Wisconsin Card Sorting Test) each appeared in 12.0% (3/25) [28,31,35]. Memory was assessed in 32.0% (8/25) [27,28,31,33,34,35,38,39] of studies (the tests included Rey–Osterrieth Complex Figure, Children’s Memory Scale, WISC Digit Span), with working memory being the most common measure (20.0%, 5/25) [27,28,31,35,38].

The majority of studies evaluating executive functions (62.5%, 5/8) [27,28,31,35,38] reported significant differences in individuals with dyslexia, particularly in working memory (WISC Digit Span) and inhibition tasks (Stroop Test). These findings suggest that dyslexia involves core challenges in executive functioning beyond reading difficulties. Spelling assessments consistently showed group differences and memory measures produced mixed results. Notably, all studies using comprehensive behavioral ratings (BRIEF) found significant executive function impairments, supporting the view that dyslexia often co-occur with broader cognitive challenges.

### 3.4. Brain Functional Connectivity (rs-fMRI)

#### 3.4.1. rs-fMRI Acquisition

Among the studies included in this review, approximately 93.7% acquired the rs-fMRI data using high-field MRI scanners (3 Tesla) [20,22,23,24,25,26,27,28,29,30,31,32,33,34,35,36,37,38,39,40,41,42,43], while only 6.2% used lower-field systems (1.5 Tesla) [21,44]. The predominant use of 3 T scanners reflects their clear advantages for rs-fMRI, such as a higher signal-to-noise ratio, enhanced spatial resolution, and greater sensitivity in detecting FC patterns. In contrast, although 1.5 T systems are more widely available and commonly used in clinical environments; they offer reduced sensitivity for research applications, requiring detailed functional brain mapping. Regarding scanner manufacturers, Philips was the most frequently reported among the 3 T studies (47.8% 11 out of 23) [22,28,29,31,34,35,37,38,41,42], followed by Siemens (43.4% 10 out of 23) [20,23,25,26,27,30,33,40,43] and GE (17.4%) [24,32,36,39], as shown in Table 2.

Notably, the two studies using 1.5 T scanner were conducted in different geographical and temporal contexts, suggesting that scanner availability and local infrastructure may influence in the selection of field strength. One study, conducted in Austria in 2014, used a Philips 1.5 T system [44], while the other, carried out in Turkey in 2024, employed a Siemens 1.5 T scanner [21] (Table 2). These cases highlight how technological evolution and resource availability over time and location can shape methodological choices in rs-fMRI research.

In terms of imaging sequences, the echo-planar imaging (EPI) sequence was used in 80% (20 out of 25) of studies [20,22,23,24,25,26,27,28,31,32,34,36,37,38,39,40,41,42,43,44], while 20% (5 out of 25) [21,29,30,33,35] reported the use of T2*-weighted sequences (Table 2). Despite being listed separately, EPI sequences are typically based on T2* contrast, which is essential for capturing the BOLD signal in functional imaging. The EPI is a fast pulse sequence that allows for whole-brain image acquisition in a matter of seconds, making it ideal for fMRI studies as it can rapidly detect subtle BOLD signal changes related to neural activity. This suggests methodological equivalence in most cases, even when sequence naming varied.

Regarding the acquisition parameters of rs-fMRI, the repetition time (TR) was predominantly set around 2000 ms, with 76.0% (19 out of 25) [20,22,23,25,26,27,31,32,33,34,35,36,37,38,39,40,41,42,43] of studies using this or a closely related value. A smaller proportion of the selected studies (16.0%) reported TR values equal to or less than 1000 ms [24,28,29,30], and two studies chose values bigger than 2000 ms [21,44]. The echo time (TE) was generally close to 30 ms, although a few studies employed slightly higher values. The combination of a TR around 2000 ms and a TE near 30 ms reflects commonly accepted parameters in rs-fMRI protocols aimed at optimizing BOLD sensitivity, indicating a notable degree of methodological consistency across studies, as shown in Table 2.

The number of slices required to acquire a full brain volume is directly related to the slice thickness, as thinner slices demand a greater number to ensure complete anatomical coverage, as reported in all the selected studies. However, only 64% (16 out of 25) [20,21,22,23,24,25,26,27,29,30,34,36,39,40,43,44] of them reported the slice numbers used (Table 2), which ranged from 25 to 140, and the slice thickness, which varied from 2 to 5 mm. Among these, most studies (8/25) used between 25 and 40 slices with a thickness of 3 to 4 mm [20,21,25,27,34,36,40,44], while others (7/25) used between 48 and 64 slices with thinner slices ranging from 2 to 3 mm [22,23,24,26,29,30,43]. Curiously, one study (1/25) used 140 slices with 3.5 mm [39], likely aiming to cover more than the entire brain.

Another important parameter influencing spatial resolution is the field of view (FOV), which was reported in 64% of the studies (16 out of 25) and ranged from 192 to 240 mm. The most frequently used FOV was 192 mm (16% 4 out of 25) [21,23,25,33], followed by 220 [24,26,36,44] and 240 mm [27,32,34,39] (each with 16%). Intermediate values, such as 200 mm [40], 210 mm [30], 211 mm [26], and 224 mm [20], were less commonly reported. Notably, 44% of studies did not report FOV values, once again emphasizing inconsistencies in methodological reporting that may affect reproducibility and hinder meaningful comparisons across studies.

The number of volumes acquired also varied considerably across studies, ranging from 75 to 857 volumes. This corresponds to acquisition times between approximately 2.5 and 10.2 min, depending on the repetition time (TR) used. In general, a higher number of volumes increases the total temporal resolution and enhances the statistical robustness of the analysis. For this reason, some studies (16%) opted to acquire multiple shorter runs and later concatenate the sequences during analysis [22,23,26,33]. Nevertheless, most of studies (32%, 8/25) preferred intermediate acquisition times ranging from 6 to 9 min (approximately 180 to 300 volumes) [24,25,27,32,36,39,40,43]. Others (24% 6/25) adopted longer single runs, typically around 10 min in duration and comprising more than 300 volumes [28,29,30,31,35,38], while a subset of 24% (6/25) employed shorter protocols lasting less than 5.5 min, acquiring between 75 and 165 volumes [21,34,37,41,42,44].

The condition under which rs-fMRI is acquired, such as eyes open, closed, or fixated, can significantly influence brain activity patterns and must be considered when interpreting FC results [47]. Most studies (44% 11/25) instructed participants to remain awake while looking at a fixation cross; one of these studies instructs the participants to let their minds wander, a widely adopted approach to minimize eye movements and maintain alertness [29,30,31,35,36,37,38,39,41,42,43]. Other studies (24% 6/25) instructed participants to stay awake with their eyes closed or simply to keep their eyes closed, not thinking about anything [20,22,23,24,25,26], while 12% (3/25) used the condition “awake with eyes open” [21,34,44]. Only one study (4%) explicitly instructed participants to remain motionless and avoid thinking [40]. Notably, 16% (4/25) of the studies did not report the specific resting-state condition, highlighting a gap in methodological transparency that may affect data interpretation and reproducibility [27,28,32,33].

#### 3.4.2. rs-fMRI Processing

Among the included studies, 56% (14/25) used the CONN toolbox software for rs-fMRI data processing, with versions ranging from 13 to 18 [26,27,28,29,30,31,32,35,37,38,39,41,42,43,44]. Earlier versions were typically associated with SPM8, while more recent ones were used in conjunction with SPM12. In 8% of the studies, other pipelines were used, including FSL (with one specifying the MELODIC tool) [21,33], SPM (version 8 or 12) combined with the GIFT toolbox [42], or AFNI (in one case, version 17.2.17) [36]. Additionally, DPABI (version 4.5 and 5.1) or DPARSF were employed for preprocessing only [20,23,34,40], also representing 8% of each. Other tools, such as BrainVoyager [24], and Dynamic BC Toolbox [23], were rarely mentioned (4%) and only one study (4%) did not mention the software used [22].

rs-fMRI preprocessing includes volume realignment, co-registration with the T1 image, atlas-based segmentation, normalization, smoothing, and noise removal. While these steps are generally standardized across software platforms, the denoising stage represents a key source of methodological variability. Studies adopted a range of thresholds to identify motion artifacts and BOLD signal fluctuations, in addition to applying temporal band-pass and spatial filters. A conservative setting was used in 32% of studies [27,30,32,33,34,38,39,44], defining outlines as volumes with framewise displacement (FD) > 0.5 mm or BOLD signal change (BSC) > 3 standard deviations (SD). Liberal thresholds (FD > 2 mm and BSC > 9 SD) were used in 20% of studies [20,21,23,25,26,37,41], while 8% applied intermediate values (e.g., FD > 0.9 mm and BSC > 5 SD) [36,43]. Notably, 28% of the studies did not report denoising criteria [22,24,28,29,31,35,40], and one study (4%) employed independent component analysis (ICA) for noise removal [42].

The most frequently used high-pass filter for temporal filtering was 0.1 Hz (32.0%) [23,26,32,33,34,36,39,43,44], followed by 0.2 Hz (24%) [28,31,35,37,38,41], 0.08 Hz (16.0%) [20,22,25,40], and 0.09 Hz (12.0%) [29,30,44]. These values indicate a general preference for low-frequency filtering around 0.1 Hz, in line with standard recommendations for rs-fMRI preprocessing.

Regarding spatial smoothing, most studies (44%) applied full width at half maximum (FWHM) of 8 mm [26,27,29,32,35,37,38,41,42,43,44], followed by 6 mm (28%) [20,22,25,33,34,36,39], 5 mm (12%) [23,24,30], and 4 mm (8%) [23,40]. In 8% of studies, smoothing parameters were not specified [28,31]. These findings reflect a general preference for moderate to high smoothing.

The most commonly used method for assessing FC among the selected studies (52% [48] 13 out of 25) was the univariate seed-to-voxel approach [24,26,27,29,30,31,34,36,39,40,43,44], which explores the correlation between a predefined seed region and all other voxels in the brain. This was followed by the bivariate ROI-to-ROI analysis (36% 9/25) [20,22,23,25,33,35,37,38,41,43], which examines connectivity between pairs of ROIs. Independent component analysis (ICA) was applied in 12% of the studies [21,28,42], while multivariate graph theory approaches were used in 8% [31,37], and 4% employed multi-voxel pattern analysis (MVPA) [27], as shown in Table 2.

Seeds and ROIs were defined based on brain segmentation using various anatomical atlases. The most commonly used reference (92.0% 23 out of 25) was the MNI (Montreal Neurological Institute) template [20,21,23,24,25,26,27,28,29,30,32,33,34,35,36,37,38,39,40,41,42,43,44]. Both the Harvard–Oxford atlas (including its use via FSL) and the WFU PickAtlas were used in 20% (5/25) of the studies [27,29,31,35,37]. Less frequently used atlases included Talairach [22], AAL (Automated Anatomical Labeling) [26], Glasser [33], Juelich Histological [39], and Haskins Pediatric Atlases [36], each reported once, in approximately 4% of the studies.

Among the selected studies, ROIs were selected from anatomical or functional atlases or derived from functional brain networks of interest, serving as seeds in seed-to-voxel analyses or as target regions in ROI-to-ROI analyses. Based on this, ROIs were grouped according to their representative functional networks. Many studies investigated more than one network, so to better represent the findings, we categorized the results into 10 functional networks: the dorsal attention network (DAN) was the most frequently examined (52%) [21,22,23,29,30,33,34,35,36,38,42,43], followed by the frontoparietal network (FPN, 44%) [20,21,22,29,30,32,35,41,42], and the visual (VIS, 36%) [20,22,23,24,25,26,34,42,44]. The cingulo-opercular network (CON) [22,29,30,32,35,37,41] and the sensorimotor network (SMN) were investigated in 28% of studies [20,21,22,23,30,42]. FPN and CON are often considered components of the executive control network. Additionally, 24% of studies analyzed the salience network (SAL) [22,26,28,31,39,43] and 20% examined the auditory network [20,22,33,40,42,44] or ventral attention network (VAN) [21,22,29,35,42]. The default mode network (DMN) appeared in 12% of the studies [21,22,27,43], while the language network was explored in 8% [22,44].

#### 3.4.3. rs-fMRI Outcomes

The outcomes were analyzed through comparisons of FC between groups and by examining correlations between connectivity results and clinical data (including demographic and neuropsychological measures). In the analysis of group comparisons regarding FC in the selected articles, we observed that 28% (7/25) of the studies reported both an increase and a decrease in FC in the dyslexia group compared to controls [20,21,27,31,32,42,44]. In contrast, 24% (6/25) of the studies reported only a reduction in FC [24,33,35,36,38,40], while only 12% (3/25) reported an increase [23,26,39]. Thus, the most frequent pattern in the dyslexia group was a decrease in FC compared to controls, especially among children, who accounted for 92% of these results.

Among the studies on adults, only two were included [43,44], and only one of them showed significant results for investigating the issue of neuroplasticity in young adults [44]. Interestingly, this study was among those that reported both an increase and a decrease in FC in the dyslexia group.

Additionally, 20% of the studies found no significant differences between groups [29,34,37,41,43], while 16% did not report or explicitly mention this information [22,25,28,30].

When analyzing the anatomical distribution of FC findings, we observed that, in dyslexia, alterations predominantly occur in ipsilateral connections (60%) [20,21,23,24,26,27,31,32,33,35,36,38,39,40,44]. Among these, 40% were located in the left hemisphere [20,21,23,24,32,33,35,36,38,39,40,44], with a higher concentration of FC reductions, while 20% were in the right hemisphere, where increased FC was more prevalent [21,24,26,27,31]. Only 16% of the studies also identified contralateral changes, involving interhemispheric communication [20,23,31,44].

Interestingly, 32% of the ipsilateral results were concentrated in the posterior region of the brain [20,21,23,33,35,36,42,44], while only 16% reported changes in connections between anterior and posterior regions, particularly along the frontoparietal pathway, where increased FC was more frequent in individuals with dyslexia [20,23,32,39]. These findings suggest that, especially in children with dyslexia, FC alterations predominantly occur in ipsilateral connections within the left hemisphere, showing a pattern of decreased connectivity, particularly in posterior brain regions.

Outcomes related to the correlation analysis between FC results and neuropsychological data were reported in 80% of the studies. Among these, 64% used reading fluency tests [22,23,24,25,28,30,31,32,34,37,39,40,41,42,43,44], 12% employed mood or emotional assessments [26,31,37], 8% included phonological awareness tasks [23,37], and smaller proportions used writing measures, RAN, and specific tests (4% each) [23,24,30,32]. However, most studies (32%) conducted the correlation analysis including both groups [22,23,24,30,31,39,40,42]. Another 12% performed the analysis based on connectivity differences between groups [34,42,43], with a focus on greater connectivity in controls compared to individuals with dyslexia, while 8% analyzed only the control group [25,34]. Additionally, 4% of the studies did not clearly specify how the analysis was conducted [44]. In the target population, participants with dyslexia, only 24% reported this correlation, which would be essential for a better understanding of how the altered connectivity patterns in this group relate to neuropsychological data [22,28,32,37,41,43].

When evaluating the descriptions of correlations in the dyslexic group, it was observed that improvements in reading fluency were associated with increased connectivity between various regions, including connections between lateral sensorimotor areas and visual regions [22] of the SAL [28], between areas of the SAL and frontal regions of the executive function network, as well as between bilateral regions related to cognition [28]. Connectivity between the left inferior parietal lobule (IPL) and the posterior cingulate cortex (PCC) was also reported [32]. However, these descriptions are relatively vague and do not allow us to conclude that these networks are specifically linked to the classical reading circuitry. Only one study clearly highlighted increased connectivity in posterior regions of the left hemisphere. Negative correlations were observed between improvements in reading fluency and decreased FC in several networks, including the FPN [37], the CON [41], subcortical regions, areas related to language, the primary visual cortex (V1) [28], as well as between the supramarginal gyrus (SMG) and angular gyrus (AG) [43]. Although these areas are not directly part of the classical reading network, they are functionally linked to executive function, language and SAL, suggesting that broader cognitive mechanisms may influence reading performance in individuals with dyslexia.

## 4. Discussion

This systematic review demonstrated that the majority of dyslexia studies assessing brain FC using rs-fMRI focused on children (92%), highlighting a significant gap in the literature regarding adults with dyslexia. This limits our understanding of brain maturation processes and neuroplasticity across the lifespan. Regarding FC findings, the results revealed that alterations predominantly occurred in ipsilateral connections (60%), with a particular emphasis on the left hemisphere (40%), where reductions in connectivity were most frequent, especially in posterior regions. This pattern is consistent with neurobiological hypotheses of dyslexia that point to dysfunctions in the phonological route and visuo-phonological integration [49]. In contrast, increased connectivity was observed in the right hemisphere (20%), possibly reflecting compensatory mechanisms.

A relevant aspect observed in this review was the predominance of studies conducted in English-speaking countries, with 52% carried out in the United States. This geographical and linguistic concentration may introduce a significant bias in the literature, limiting the generalizability of findings to populations from other regions. However, a recent bibliometric analysis identified 9166 publications on dyslexia between 2000 and 2021, indexed in the SSCI and SCI-E databases, highlighting that the United States, the United Kingdom, and Germany were the three most productive countries, while China, Israel, and Japan led dyslexia-related publications in the Asian context. Moreover, the University of Oxford stood out as the institution with the highest number of publications and the highest h-index in the field of dyslexia [50].

Additionally, a predominance of studies conducted in opaque languages (such as English and Chinese) was noted, accounting for 64% of the included articles. These languages feature more complex and irregular phoneme–grapheme correspondence rules, which tend to intensify the symptoms of dyslexia [51]. In contrast, in transparent languages such as Italian, Spanish, and Portuguese, where the correspondence between letters and sounds is more regular, individuals with dyslexia typically exhibit difficulties more related to reading speed than accuracy. This difference may reflect distinct phenotypic profiles of dyslexia [52]. Therefore, the predominance of studies in opaque languages may overestimate or distort specific characteristics of dyslexia that do not manifest in the same way in more transparent linguistic contexts.

This geolinguistic asymmetry highlights the need to promote greater diversity in sample populations and linguistic contexts analyzed, in order to enhance the representativeness of neurofunctional findings and advance the global understanding of dyslexia.

In demographic terms, there was a predominance of samples composed of male participants, reflecting the higher prevalence of dyslexia among males. Dyslexia is a condition known to exhibit gender differences, with a greater incidence in males [6], a pattern that was generally mirrored in the selected studies. Additionally, sample sizes varied considerably across studies and were generally modest, an aspect that may limit the statistical power and robustness of the FC analyses.

Interestingly, among the studies included in this review, only two investigated adults with developmental dyslexia, and one of them included exclusively male participants [44]. This limited representation highlights a critical gap in the literature and underscores the importance of understanding how the adult brain adapts to dyslexia over time, particularly considering the long-term effects on educational attainment, professional opportunities, occupational trajectory, and socioeconomic independence. Such insights are particularly relevant for individuals who carry significant family responsibilities, such as heads of household. One study explicitly reported that socioeconomic status plays a significant role in access to dyslexia diagnosis, and in shaping educational and employment experiences during adulthood [53]. These findings emphasize the need to examine dyslexia not only through a neurobiological lens but also considering contextual and socioeconomic factors that may modulate outcomes in adult populations.

As is often the case in the literature, the majority of functional neuroimaging studies in dyslexia focus on children, with adult data being markedly scarce, a limitation also reflected in this review [11]. Regarding children, findings are often inconsistent, largely due to methodological variability in functional connectivity analysis, including differences in the networks and regions of interest (ROIs). Nevertheless, multiple studies [20,21,22,23,24,25,26,27,28,29,30,31,32,33,34,35,36,37,38,39,40,41,42] report that dyslexic children, compared to age-matched controls, exhibit altered functional connectivity in reading-related networks, such as the dorsal sensorimotor, frontoparietal, visual, and auditory networks, often showing increased connectivity in cognitive control regions and decreased connectivity in core language and subcortical areas. In adults, few studies are available [43,44] for comparison, which makes any generalizability difficult; the limited findings for adults generally align with those from pediatric populations, revealing abnormal connectivity patterns within reading networks, as well as the compensatory recruitment of executive control networks, such as the frontoparietal and salience networks. However, due to a significant gap in adult representation, the only conclusion plausible at this point is about the urgent need for more functional studies to clarify the developmental trajectory and long-term neural adaptations associated with dyslexia.

It is important to recognize that most neuropsychological diagnostic tools are inherently language specific and require rigorous adaptation and validation to align with the linguistic and cultural context of each population [54]. The heterogeneity of instruments observed across the included studies reflects this necessity, particularly in cross-cultural neuropsychology, where constructs such as “phonological awareness” manifest differently depending on the language’s orthographic transparency. For instance, pseudoword reading tasks like the TOWRE are widely used and valid for assessing phonological deficits in opaque languages such as English, but they show limited discriminative power when applied to logographic languages like Chinese. In such cases, tools like the Chinese Reading Test Battery (CRTB), which emphasizes character recognition, are more ecologically valid [55].

In the present review, the neuropsychological tests analyzed were those reported in each included study to assess dyslexia and to identify the presence or absence of relevant comorbidities, such as ADHD, dyscalculia, and dyspraxia. This justified our decision to review and describe these instruments, given their role in characterizing participant profiles and controlling for overlapping cognitive impairments. Consequently, the diversity of tests reflects not only linguistic differences but also variations in the scope of neuropsychological screening conducted across studies. It was therefore relevant to evaluate how each study assessed key cognitive domains related to the reading process and its associated functions.

Although this variability poses challenges for direct comparisons of cognitive impairments within dyslexia across studies, it does not necessarily compromise the synthesis of findings. Some studies have shown that core deficits in dyslexia, such as impairments in phonological processing and rapid automatized naming, can be reliably identified using appropriately adapted tools [56]. Nonetheless, we acknowledge that due to the lack of cross-cultural protocols for dyslexia assessment, the account for nuances of the interaction between the development and reading acquisition in different linguistic contexts and must be considered when interpreting cross-study patterns and drawing generalizable conclusions. A critical evaluation of test validity and cultural applicability is therefore essential in future reviews and meta-analyses.

Regarding the FC findings, most studies used high-field MRI scanners (3T), which represent an important methodological advantage. The increased magnetic field strength improves the signal-to-noise ratio (SNR), allowing for a more precise detection of spontaneous brain activity fluctuations during rest [57]. Additionally, higher spatial resolution (with slice thicknesses under 3 mm) [24,25,27,43] and temporal resolution (TR < 2000 ms) [23,25,27,29,30,31,43] enable a more detailed identification of neural networks, including small and deep brain regions. This combination enhances the robustness and reproducibility of the findings, increasing the reliability of FC pattern interpretations and aligning with the conservative parameters adopted in most studies during data processing.

In addition to the methodological heterogeneity in analytical approaches, this review also revealed substantial variability in imaging acquisition protocols across the included studies, which may influence reproducibility and the comparability of findings. Although most studies reported basic acquisition parameters, only a portion provided full details regarding head motion thresholds, scan duration, and participant instructions. Notably, most studies adhered to recommended standards for resting-state fMRI acquisition, employing total scan times exceeding six minutes, a critical threshold to ensure reliable functional connectivity estimates [48]. Some studies adopted a strategy of performing multiple shorter acquisitions, later concatenated into a single sequence, to improve data quality and minimize motion artifacts. In line with established guidelines, the majority of studies instructed participants to keep their eyes open with visual fixation (typically on a crosshair), while others used eyes closed or eyes open without fixation protocols, which can differentially impact baseline network activity [58]. Temporal resolutions of 2 s or less were also commonly applied, aligning with best practices for capturing dynamic fluctuations in brain activity during resting state [48].

In terms of preprocessing procedures, we observed additional heterogeneity. To better interpret this, we classified the denoising strategies used by the studies as conservative (32%), intermediate (8%), or liberal (20%), based on the presence and extent of motion correction, physiological noise removal, and filtering methods. These differences in preprocessing pipelines can significantly affect functional connectivity results and should be more consistently reported in future studies to support replication and synthesis. We recommend the adoption of standardized reporting practices, such as those proposed by recent neuroimaging guidelines, to enhance transparency and comparability in this growing field [47].

All connectivity methods applied in the analyzed studies were based on traditional static FC approaches, with the exception of one study that employed effective connectivity analysis [20]. Regarding the statistical approaches, most studies used bivariate methods such as seed-to-voxel, ROI-to-ROI, and Granger causality analysis, which assess the relationship between pairs of brain regions. Notably, the predominance of seed-to-voxel studies [24,26,27,29,30,31,34,36,39,40,43,44] reflects the exploratory nature of this research, given the lack of clarity and consistencies regarding which neural networks are functionally altered in dyslexia. Less frequently, multivariate methods such as graph theory, independent component analysis (ICA), and multi-voxel pattern analysis (MVPA) were used, allowing for the examination of more complex spatial patterns and simultaneous interactions across multiple brain regions [59,60].

The neural networks DAN, FPN, CON, SMN, and SAL were frequently reported as ROIs in the selected studies, despite not being part of the classical reading circuit. The VIS network, which is directly involved in word and letter recognition, was also widely reported. These networks play fundamental roles in modulating auxiliary cognitive processes that are essential for fluent and effective reading: the DAN is associated with visual and spatial attention; the FPN with executive control, decision making, and working memory; the SAL with salience detection and the prioritization of relevant linguistic information; the CON with sustained attention and error monitoring; and the SMN with motor coordination and sensorimotor/visual integration [61].

Importantly, this interaction among these networks has led to their functional classification into three main hubs, which integrate information across distinct neural systems, despite occupying different structural positions and engaging differently depending on the task. The first, referred to as the ’default-control connector hub’, involves the FPN, DMN, and contextual association networks (CANs). The second, known as the ’cross-control connector hub’, is composed of the CON, DAN, and FPN networks, mediating the transition between internal and external attention. The third, the ’processing-control connector hub’, integrates sensory and motor networks (VIS, auditory, and somatomotor) with CON and DAN, coordinating sensory and motor processing with attention and control [62].

In addition to these frequently reported networks, other systems that are equally important for reading, either directly or in a complementary manner, were also reported, although less consistently across studies. These include the auditory network, essential for phonological processing; VAN, involved in the reorientation of attention to unexpected stimuli; DMN, which may interact with attentional systems during reading; and the language network, classically associated with phonological and semantic processing [61]. In individuals with dyslexia, altered FC within these networks may reflect either compensatory mechanisms or additional dysfunctions that negatively affect reading performance [13].

A consistent finding across studies was a widespread pattern of decreased FC, particularly in the left and posterior regions of the brain, areas typically implicated in reading-related processes. The most frequently reported regions with reduced activity included the left MTG, inferior frontal gyrus (IFG) (especially the pars triangularis), frontopolar cortex, and ACC. Additional reductions were observed in limbic and temporal structures such as the amygdala, hippocampus, MTG, ITG, and STG. In the parietal lobe, decreased signal was consistently found in the precuneus, angular gyrus, and SMG, indicating potential disruptions in attentional control and sensory integration.

At the network level, hypoconnectivity was reported in key systems involved in cognitive control and self-regulation, including the FPN, DMN, and CON. Reductions were also observed in visual processing areas such as the calcarine cortex, occipital lobe, and the VWFA, reinforcing the notion that visual decoding is impaired in dyslexia [63]. These findings align with evidence showing that dyslexia is marked by a disruption in posterior left-hemispheric networks during phonological processing [64], and that the development of visual tuning for print, especially within the inferior occipitotemporal regions, is delayed in children with dyslexia [65]. Together, this widespread hypoconnectivity across cognitive, sensory, and language-related networks underscores the multifactorial nature of reading difficulties in dyslexia.

In contrast, increased FC was reported less frequently and appeared more regionally specific. Notable examples included connections between the cuneus and the IFG, as well as between the right anterior insula (vAI) and subcortical structures such as the amygdala, thalamus, hypothalamus, and brainstem. These findings may reflect the brain’s attempt to recruit alternative circuits to compensate for deficits in regions traditionally involved in reading, such as the visual word form area (VWFA), superior temporal gyrus (STG), and inferior parietal lobule (IPL) [63]. Increased connectivity with frontal regions may also indicate the greater engagement of executive control and attentional systems in response to difficulties with automated phonological decoding [66,67]. The bilateral amygdala showed increased connectivity with the middle frontal gyrus (MFG) and the right temporal pole (TP), suggesting the recruitment of emotional and subcortical networks. This pattern may be linked to greater emotional load during reading, potentially reflecting anxiety, frustration, or heightened attentional vigilance in response to the task [68]. Additional increases in connectivity were observed involving the medial prefrontal cortex (MPFC), DMN, precuneus, and the left fusiform gyrus (FFG). Specifically, increased connectivity in the left fusiform region may represent an effort to enhance VWFA activation, which is critical for visual word recognition. The recruitment of these other regions, although atypical, may serve as alternative support mechanisms for reading processes in children and adults with dyslexia.

Correlations between FC and reading skills were the most frequently reported outcomes in the selected studies, highlighting the relevance of reading performance in understanding the neural patterns associated with dyslexia. Positive correlations were predominantly found in regions such as the left IPL, superior medial gyrus (SMd), medial prefrontal cortex (mPFC), as well as the inferior temporal gyrus (ITG) and superior temporal gyrus (STG). These areas are involved in grapheme–phoneme conversion, cognitive control, and phonological and semantic processing [69]. The findings suggest that stronger connectivity in these regions may reflect effective compensatory strategies in individuals with better reading performance.

In contrast, negative correlations were reported less frequently, involving primarily the CON, left IFG, left SPL, and the superior and middle frontal gyri (SFG and MFG). In these cases, increased connectivity was associated with poorer reading performance, possibly indicating excessive cognitive effort, attentional overload, or atypical recruitment of regions that would typically support more automatic reading processes. These findings align with evidence of functional disruptions in specific regions among dyslexic individuals, such as the posterior left middle temporal gyrus (post-LMTG), which although often overlooked in studies involving typical readers has been consistently found to be hypoactive in dyslexic readers and responsive to intervention [70].

### Limitations

One of the challenges identified in this systematic review was the wide variability in terminology used to refer to dyslexia, such as ’reading disorder’, ’reading disability’, ’learning impairment’, among other variations. This terminological inconsistency can negatively impact the identification and selection of relevant studies, compromising the comprehensiveness and accuracy of systematic reviews and hindering clear comparative analysis across different studies. In addition, substantial variability was also observed in the methodologies used to assess FC, as well as in the selection of neural networks and ROIs, many of which were not directly related to the reading process. These inconsistencies further complicate the integration and interpretation of findings across studies, limiting the development of a coherent neurobiological model of dyslexia.

Another relevant limitation was the frequent absence of detailed information regarding participants’ educational background or reading age. Although most studies used neuropsychological assessments adjusted for age and/or schooling, only a few explicitly reported these variables [20,21,25,27,33]. For instance, the two studies involving adults with dyslexia did not clarify whether participants had completed compulsory education, which is critical for interpreting long-term compensatory mechanisms. Given that reading experience and educational attainment significantly influence brain plasticity, the omission of such data weakens the developmental interpretation of rs-FC alterations and highlights the need for more comprehensive and standardized reporting in future studies.

## 5. Conclusions

This systematic review revealed a predominance of studies involving children, highlighting functional hypoconnectivity in posterior regions of the left hemisphere in dyslexia, consistent with phonological pathway deficits. Increases in connectivity were less frequent and possibly compensatory. Networks such as DAN, FPN, CON, and VIS were frequently involved, indicating the importance of auxiliary processes like attention and executive control. Correlations between FC and reading skills reinforce the complexity of the neural mechanisms underlying the disorder. Methodological heterogeneity and the focus on opaque languages limit generalizability, emphasizing the need for more diverse and standardized studies.

## Figures and Tables

**Figure 1 biomedicines-13-01210-f001:**
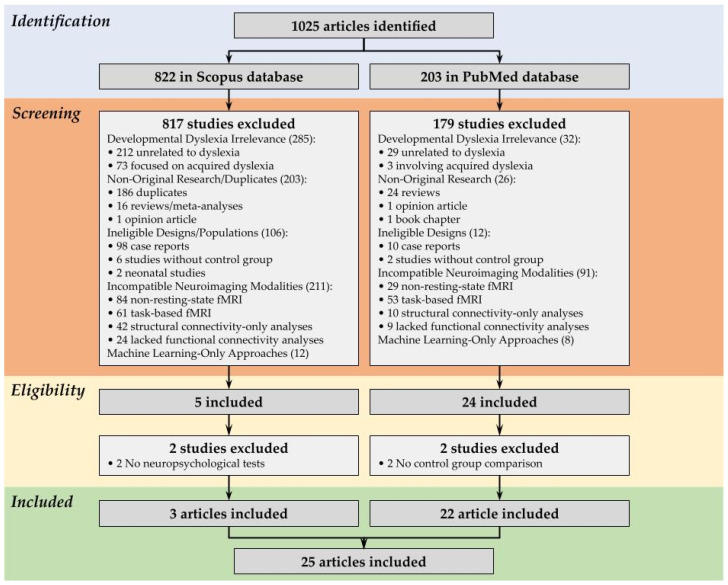
The PRISMA flowchart of the systematic review showing the inclusion and exclusion of studies.

**Figure 2 biomedicines-13-01210-f002:**
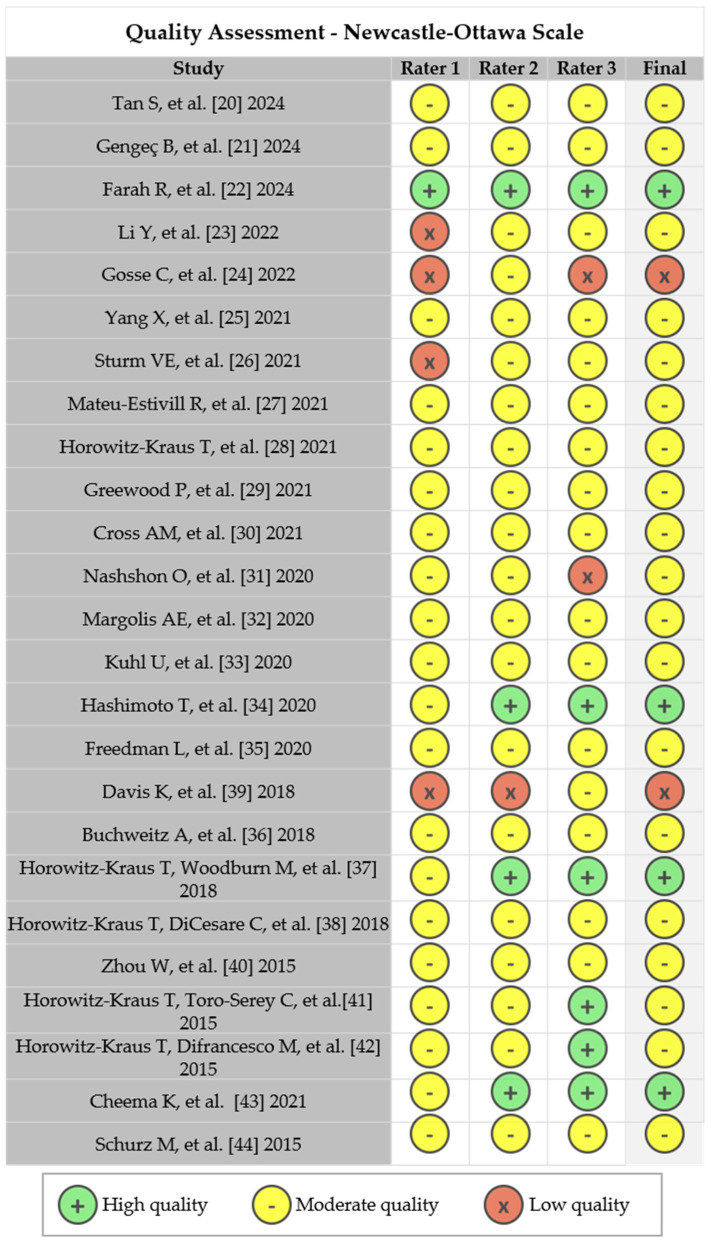
Quality assessment using the Newcastle–Ottawa Scale (NOS) for each study [20,21,22,23,24,25,26,27,28,29,30,31,32,33,34,35,36,37,38,39,40,41,42,43,44].

**Table 1 biomedicines-13-01210-t001:** Demographic characteristics of dyslexia of development.

Ref.	Year	Demographic Data							Neuropsychological Tests for Dyslexia Evaluation							
		Country	Language	Group	N	Sex W:M	Age (Years)	Years of Education (or Level)	IQ	Word Reading Accuracy	Pseudoword Reading Accuracy	Text Reading Accuracy	RAN	Phonological Task	Attention	Other
rs-fMRI in Children																
Tan S et al. [20]	2024	China	Chinese	Control DYS	36 64	8:28 25:39	9.4 ± 1.1 9.8 ± 1.1	2nd to 5th grade	WISC-IV (FSIQ ≥ 80)	**CRTB**	NR	NR	**Digits, Letters, Colors and Shapes**	NR	Conners Teacher Rating Scale (CBRS)	**PRS**
Gengeç B et al. [21]	2024	Turkey	Turkish	Control DYS DYSe	15 13 15	10:5 3:10 6:9	10.1 ± 1.5 8.9 ± 1.7 9.4 ± 1.5	E-DYS: DYS with special education for 12–24 months	**WISC-IV (≥80)**	**Number of words read per minute**	NR	NR	NR	NR	NR	Motor coordination assessment
Farah R et al. [22]	2024	USA	English	Control DYS	65 55	29:36 24:31	10.1 ± 1.7 9.9 ± 1.3	NR	Non-verbal Intelligence Test (TONI)	**Letter–Word and Word Attack subtests (WJ-III);** **Sight Word Efficiency subtest (TOWRE)**	**Timed pseudoword decoding efficiency (TOWRE)**	NR	NR	**Elision (CTOPP)**	NR	Stroop, Stories listening
Li Y et al. [23]	2022	China	Chinese	Control DYS	38 22	17:21 17:5	10.9 ± 0.8 10.7 ± 0.9	NR	Non-verbal (Raven)	NR	NR	**Reading Fluency Test**	NR	**Oddity detection task**	NR	**Vocabulary (CRM)**
Gosse C et al. [24]	2022	Belgium	French	Control DYS	16 16	9:5 5:9	9.3 ± 1.0 9.4 ± 1.2	NR	Non-verbal (WISC-IV)	**Analytic Battery of Written Language (Regular and Irregular Words)**	**Analytic Battery of Written Language (Nonwords)**	NR	NR	NR	NR	**Spelling (BALE)**, Concise Evaluation Scale for Children’s Handwriting
Yang X et al. [25]	2021	China	Chinese	Control DYS	25 18	13:12 6:12	10.4 ± 1.0 10.2 ± 1.0	10:12 (4°:6°)	Non-verbal (Raven)	**CRTB**	NR	**Reading Fluency Test**	NR	NR	NR	NR
Sturm VE et al. [26]	2021	USA	English	Control DYS	22 32	11:11 15:17	10.5 ± 1.5 10.3 ± 1.4	NR	Non-verbal (WASI)	**Sight Word Efficiency (TOWRE-2)** and Letter–Word (WJ-IV)	**Phonemic Decoding Efficiency (TOWRE-2)** and Word Attack (WJ-IV)	Paragraph Reading (GORT-5)	NR	NR	NR	Social Skills, Anxiety and Depression (BASC-2)
Mateu-Estivill R et al. [27]	2021	Spain	Catalan	Control DYS	30 34	NR	7.4 ± 0.5 7.1 ± 0.7	1st or 2nd grade	VIQ and PIQ (WISC-IV)	**Words subtest (TALE-C)**	**Pseudowords subtest (PROLEC-R)**	**Text subtests (TALE-C)**	Letters, **Objects** and **Colors**	NR	NR	**Executive Function and Verbal Fluency (FAS, Animals), Spelling (TALE-C),** Digit span (WISC-IV)
Horowitz-Kraus T et al. [28]	2021	USA	English	Control DYS	28 29	NR	10 ± 1.3 9.8 ± 1.3	NR	Non-verbal Intelligence test (TONI-3) and VIQ (PPVT-4)	**TOSREC**, **Sight Word Efficiency (TOWRE)** and **Letter–Word (WJ)**	**Pseudoword Efficiency (TOWRE)** and **Word Attack (WJ)**	**Passage Comprehension (WJ)**	**Numbers, Objects, Colors and Letters (CTOPP)**	**Elision (CTOPP)**	Conners Comprehensive Behavior Rating Scales (CBRS), **Sky Search (TEA-Ch)**, Sustained (CPT)	**Working memory (Digit Span—WISC**), Coding and Symbol Search (WISC), **Switching (Stroop)**, **Fluency (DKEFS**), EF (BRIEF)
Greenwood P et al. [29]	2021	USA	English	Control DYS	21 24	6:15 12:12	10.0 ± 1.5 9.8 ± 1.4	NR	**Non-verbal Intelligence test** **(TONI-IV)**	**Sight Word Efficiency (TOWRE)** and **Letter–Word (WJ)**	**Pseudoword Efficiency (TOWRE)** and **Word Attack (WJ)**	**TOSREC**, **Passage Comprehension (WJ)** and **GORT**	NR	**Elision** and **Blending Words (CTOPP-2)**	Conners Self-report Test (CBRS)	NR
Cross AM et al. [30]	2021	Canada	English	Control DYS	65 18	45:38 (Total)	10.9 ± 1.0	NR	Non-verbal (WASI-II)	Sight Word Efficiency (TOWRE-2)	**Phonemic Decoding Efficiency (TOWRE-2)**	**Passage Comprehension** **(WJ-III)**	**Letters**	NR	NR	NR
Nachshon O et al. [31]	2020	USA	English	Control DYS	37 27	15:22 15:12	9.9 ± 1.0 10.7 ± 1.1	NR	Non-verbal Intelligence test (TONI-3) and VIQ (PPVT-4)	**Sight Word Efficiency (TOWRE)** and **Letter–Word (WJ)**	**Pseudowords Reading Efficiency (TOWRE)** and **Pseudoword Reading (WJ)**	**Passage Comprehension (WJ)**	**Numbers** and **Letters** (CTOPP)	**Elision (CTOPP)**	Sky Search (TEA-Ch)	**Inhibition (STROOP)**, **Working memory (Digit Span—WISC), Symbol Search (WISC),** Switching (Wisconsin, EF (BRIEF), Coding (WISC)
Margolis AE et al. [32]	2020	USA	English	Control DYS	18 20	10:8 8:12	10.4 ± 1.7 9.8 ± 1.3	NR	**FIQ**, **VIQ** and PIQ (WASI)	**TOWRE-2** and **WJ-III**	**TOWRE-2**	**GORT-5**	NR	NR	NR	Rapid Simon Task
Kuhl U et al. [33]	2020 (pub) 2012–2013 (initial data) 2015–2015 (final data)	Germany	German	Control DYS	16 16	7:9 5:11	BL: 5.6 ± 4 5.8 ± 4 AL: 8.8 ± 3 8.4 ± 3	BL: Kindergarten (before literacy) AL: Second grade	BL: **Non-verbal (WPPSI-III)** AL: Non-verbal (WISC-IV)	BL: - **AL:** **SLRT-II**	NR	NR	BL: Colored Objects (BISC) AL: -	BL: **BISC** AL: Base competence for school years **(BAKO)**	NR	BL: **Short-term memory (K-ABC)** AL: **German spelling test (DERET 1-2)** and Heidelberg math’s test (HRT1-4)
Hashimoto T et al. [34]	2020	Japan	Japanese (syllabic Hiragana and logographic Kanji)	Control DYS	46 22	6:40 3:19	10.5 10.7	NR	**FSIQ**, **VIQ** and PIQ (WISC-III or IV)	**Standard comprehension test of abstract words**	NR	NR	**Type not specified**	**Backward word repetition (3 mola and 4 mola word**)	NR	Rey–Osterrieth Complex Figure (copy, immediate recall and delayed recall)
Freedman L et al. [35]	2020	USA	English	Control DYS	30 26	16:14 10:16	10.0 ± 1.4 10.0 ± 1.3	NR	Non-verbal Intelligence test (TONI-3)	**Letter–Word (WJ-III)** and **Sight Word Efficiency (TOWRE)**	**Word Attack (WJ-III)**	NR	NR	NR	**Visual attention (TEA-Ch)**	**Digit span (WISC-III), Switching (D-KEFS), Wisconsin Card Sorting Task, Inhibition (BRIEF)**
Davis K et al. [39]	2018	USA	English	Control DYS	21 24	NR	10.2 ± 1.8 9.8 ± 1.2	NR	VIQ and non-verbal (WASI) and WISC-IV	Letter–Word and spelling (WJ-III)	Word Attack (WJ-III) and Phonemic Decoding Efficiency (TOWRE-2)	**Reading Fluency (WJ-III) and GORT-5**	Letters and Digits (CTOPP-2)	NR	NR	Children’s memory scale (CMS) story memory, word pairs and numbers; RCMAS, CDRS-R
Buchweitz A et al. [36]	2018 (pub) 2014–15 (tests)	Brazil	Brazilian Portuguese	Control DYS	16 16	7:9 5:11	8.4 ± 0.5 9.6 ± 0.8	NR	WASI	Word Reading Task (LPI)	Pseudoword Reading Task (LPI)	**Reading Comprehension test**	NR	NR	NR	NR
Horowitz-Kraus T, Wood et al. [37]	2018	USA	English	Control DYS MD BD	17 15 14 11	9:8 7:8 11:3 3:9	9.8 ± 1.4 10.3 ± 1.5 15.5 ± 1.8 16.7 ± 1.8	NR	NR	**Letter–Word** **(WJ-III)**	**Word Attack (WJ-III)**	NR	NR	**Elision (CTOPP)**	NR	NR
Horowitz-Kraus T, DiCesare et al. [38]	2018	USA	English	Control DYS	9 10	4:5 2:8	10.2 ± 0.7 9.9 ± 0.9	NR	Non-verbal IQ (TONI-3) and **VIQ (PPVT-4)**	**Letter–Word** **(WJ-III)**	**Word Attack (WJ-III)**	**TOSREC**	NR	**Elision (CTOPP)**	Conners Comprehensive Behavior Rating Scales (CBRS)	**Shifting (Color-word Stroop)**, Visual attention (Sky Search), **Working memory (Digit-span)**, and **Learning (Wisconsin)**
Zhou W et al. [40]	2015	China	Chinese	Control DYS	26 21	16:10 9:12	12.0 ± 1.2 12.0 ± 1.6	NR	Non-verbal (C-WISC and Raven)	NR	NR	**Reading Fluency Test**	**RAN**	**Phoneme deletion** and **Lexical decision**	NR	**Morphological production**
Horowitz-Kraus T, Toro-Serey C et al. [41]	2015	USA	English	Control DYS	17 15	8:9 8:7	9.8 ± 1.4 10.3 ± 1.5	NR	Non-verbal Intelligence test (TONI-3)	**TOWRE**	**TOWRE**	NR	**Objects (CTOPP)**	**Elision (CTOPP)**	Conners Comprehensive Behavior Rating Scales (CBRS) and Speed and accuracy (TEA-Ch)	Fluency abilities (D-KEFS), **Switching (WCST)**, **Inhibition abilities (Stroop—D-KEFS)**
Horowitz-Kraus T, Difrancesco M et al. [42]	2015	USA	English	Control DYS	18 18	9:9 9:9	9.8 ± 1.7 9.9 ± 1.3	NR	Non-verbal Intelligence test (TONI-3)	**TOWRE**	**Decoding subtest for nonwords reading (TOWRE)**	**Rate, Accuracy (GORT-IV)**	NR	NR	Conners Comprehensive Behavior Rating Scales (CBRS)	NR
rs-fMRI in adults																
Cheema K et al. [43]	2021	Canada	English	Control DYS	19 14	14:5 10:4	21.6 ± 2.0 24.4 ± 5.4	NR	Non-verbal (WASI)	**Sight Word Efficiency (TOWRE-1)** and **Word Identification**	**Pseudoword Decoding Efficiency (TOWRE-1)** and **Word Attack (WJ)**	NR	NR	NR	NR	**Spelling**
Schurz M et al. [44]	2015	Germany	German	Control DYS	14 15	0:14 0:15	17.8 ± 1.2 18.3 ± 1.1	NR	**VIQ, PIQ** (WAIS-R)	**Word Reading**	**Non-word Reading**	**Sentence Reading**	**Objects**	**Lexical decision task**	NR	NR

**Note:** Bold font indicates significant group differences. **Abbreviations:** NR: not reported; N: number of subjects; m: months; W: women; M: male; IQ: intelligence quotient; RAN: rapid automatized naming tasks; DYS: dyslexic; DYSe: dyslexics who received special education; BL: before literacy; AL: after literacy; FSIQ: full-scale intelligence quotient; TONI: test of nonverbal intelligence; FIQ: full-4 IQ; PIQ: performance IQ; VIQ: verbal IQ; CRTB: Chinese Character Recognition Test Battery and Assessment Scale for Primary School Children; TALE-C: Test d’análisis de Lectura i escriptura en català (reading–writing analysis test Catalan version); TOSREC: Test of silent reading efficiency and comprehension; LPI: Language Proficiency Index; PROLEC-R: Batería de evaluacion de los processos lectores (assessment tests of reading process); FAS: F-A-S phonemic verbal fluency test; GORT: Gray Oral Reading Test; CRM: Character Recognition Measures and Assessment Scale; CBRS: Conners Comprehensive Behaviour Rating Scale; TEA-Ch: Test of everyday attention for children; CPT: Continuous performance task; PRS: pupil rating scale; BASC: Behaviou Assessment System for children; D-KEFS: Delis–Kaplan Executive function system; EF: executive function; BRIEF: Behaviour Rating Inventory of Executive Function; CMS: children’s memory scale; RCMAS: Revised children’s manifest anxiety scale; CDRS-R: Children’s depression rating scale; WCST: Wisconsin card sorting test; TOWRE: Test of Word Reading Efficiency; SWE: Sight Word Efficiency (TOWRE subtest); CTOPP: Comprehensive Test of Phonological Processing; MD: Mood disorders; BD: Behavioral disorders; WISC: Wechsler Intelligence Scale for Children; BALE: Analytic Battery of Written Language; WJ-III: Woodcock-Johnson III Tests of Achievement; WJ-IV: Woodcock-Johnson IV Tests of Achievement; BISC: Bielefelder Screening zur Früherkennung von Lese-Reschtschreibschwierigkeiten (Bielefeld screening of literacy precursor abilities); K-ABC: Kaufman Assessment Battery for Children. German version; WPPSI-III: Wechsler Preschool and Primary Scale of Intelligence. German Version; SLRT-II: Lese- und Rechtschreibtest. Weiterentwicklung des Salzburger Lese- und Rechtschreibtests (Salzburg test of reading and spelling); DERET 1-2: Deutscher Rechtschreibtest für das erste und zweite Schuljahr (German spelling test); BAKO 1-4: Basiskompetenzen für Lese Rechtschreibleistungen (test of basic reading and spelling skills); HRT 1-4: Heidelberger Rechentest—Erfassung mathematischer Basiskompetenzen im Grundschulalter (standardized arithmetic efficiency test; PPVT: Peabody Picture Vocabulary Test; WASI: Wechsler Abbreviated Scale of Intelligence.

**Table 2 biomedicines-13-01210-t002:** Brain functional connectivity characteristics on acquisition, process, and outcomes of dyslexia.

Ref.	rs-fMRI Acquisition								rs-fMRI Process							rs-fMRI Outcomes	
	Equipment Field	Sequence	TR/TE (ms)	Slice Number	Slice Thickness (mm)	FOV	Volumes (Time)	Condition	Software Version	Denoising Process			Atlas	Type of Analyze	ROIs	Difference in FC of Groups	Clinical-Connectivity Correlation
										Outlier Parameters	Temporal Filter (Hz)	Spatial Smoothed * (mm)					
rs-fMRI in Children																	
Tan S et al. [20]	Siemens 3 T	EPI	2000/30	32	3.5	224	NR	Awake with eyes closed	DPABI 5.1	>3 mm 3° rotation	0.01–0.08	6	MNI	ROI-to-ROI (Granger causality analysis)	Left SOG, PreCG, IFG tri, MFG, SMG, SMA, PG, right PG, bilaterally MOG, and STG	DYS showed decreased EC from left PreCG to right PreCu and AG, and left ACC and calcarine, as well as from left MFG to left calcarine, compared to controls. Enhanced EC from right Cu to left IFG tri (DYS > controls).	Letter-RAN scores positively correlated with EC from left PG to left ACC.
Gengeç B et al. [21]	Siemens 1.5 T	T2*	2800/53	25	5	192	133	Awake eyes open	FSL MELODIC 6.0.4	>2 mm or mean of 0.5 mm	0.01	5	MNI	ICA AROMA-dual regression	VIS: right occipital lobe sub-gyral; DMN: bilateral MTG; FPN: l-STG, l-MTG, l-IPL, right cerebellum cerebellar tonsil, right PreCu; VAN: bilateral insula; OFN: r-ACC; LMN: l-IPL	VIS: DYS decreased FC in the right compared to EDYS and in the left MOG in EDYS compared to controls Compared to controls: DMN: DYS decreases FC in the left MTG; FPN: DYS increases FC in the STG; VAN: DYS increases FC in the left IA OFN: DYS decreases FC in the right dACC; LMN: DYS decreases FC in the left IPL	NR
Farah R et al. [22]	Philips 3 T (dataset 1 and 2)/Philips 3T (dataset 3)	EPI (dataset 1 and 2)/EPI multi-band (dataset 3)	2000/30 (dataset 1 and 2)/700/30 (dataset 3)	37 (dataset 1 and 2)/48 (dataset 3)	4 (dataset 1)/3 (dataset 2 and 3)	NR	180/300/857	Eyes closed	NR	NR	0.009–0.08	6	Talairach	ROI-to-ROI	SMd, SMl, CON, auditory, DMN, PMN, VIS, FPN, salience, VAN, DAN, MTL, BG, THA and cerebellum	No group differences	Reading fluency shows positive correlation with the FC in DYS within the SMd, FPN, and between the SMd and auditory networks for both groups, as well as between pseudoword reading fluency and FC in DYS within the SMl and VIS
Li Y et al. [23]	Siemens 3T	EPI	2000/31.8|2000/30	54|32	2|3	192	160 | 240	Awake with eyes closed, not thinking about anything	DPABI 4.5 and SPM 8 Dynamic BC toolbox	>3 mm or 3° rotation	0.01–0.1	4	MNI	Dynamic FC by ROI-to-ROI and ALFF and ReHo	Reading network: PreCG, PostCG, MFG, IFG, AG, IPL, STG, MTG, ITG, FG, MOG, and Cu bilaterally	DYS showed a greater temporal variability of FC at left IFG with the left MTG, and the left MOG with right than controls. DYS showed reduced ALFF and ReHo compared to controls.	Temporal variability in the left IFG negatively correlated with reading fluency and phonological awareness for both groups.
Gosse C et al. [24]	GE 3T	EPI	1500/30	64	2	220	250	Awake with eyes closed	Brain Voyager 20.4	NR	0.005	5	MNI	Seed-to-voxel	Left GMFA located in BA6 in the posterior part of the MFG.	DYS reduction in FC at bilateral frontoparietal, limbic, and cerebellum, as well as the left temporolimbic and temporal areas compared to controls	Reading and handwriting scores correlated positively in almost all clusters in both groups
Yang X et al. [25]	Siemens 3T	EPI	2400/30	40	3	192	180	Awake with eyes closed	SPM 12	FD > 2 mm; 2° rotation	0.01–0.08	6	MNI	ROI-to-ROI	V1, BA17	NR	Positive correlation between FC (V1-MFG) and reading fluency in controls
Sturm VE et al. [26]	Siemens 3T	EPI	2000/31| 1290/32.4	31|68	3.5|2.2	220|211	180|279	Awake with eyes closed	SPM 12|CONN 17f	<2 mm	0.01–0.1	8	MNI | AAL	Seed-to-voxel	SAL network: right ventral AI, bilateral ACC	DYS showed stronger FC in right vAI with the pACC, the anterior MCC, amygdala, THA, hypothalamus, and brainstem	Positive correlation between FC of the right vAI—pACC, anterior MCC, and the right frontal pole with emotional facial behavior.
Mateu-Estivill R et al. [27]	Siemens 3T	EPI	2500/29	40	3	240	242	NR	SPM12|CONN	FD > 0.5 mm or SIC > 3 SD	0.008–0.1	8	MNI|Harvard–Oxford Atlas	MVPA | Seed-to-voxel	PCA —Cluster 1: PCC and right MFG, —Cluster 2: the left FFG	DYS higher FC right MFG with DMN and lower with bilateral AI and SMG	No significant correlations were found
Horowitz-Kraus T et al. [28]	Philips 3T	EPI	700/30	NR	3	NR	857	NR	SPM 12|CONN 17f	NR	0.008–0.2	NR	MNI	ICA	SAL network: ACC, bilateral AI, bilateral rostral PFC; bilateral SMG	NR	DYS showed positive correlations between screen vs. reading time ratio and FC of SAL network and frontal EF regions, as well as between SAL network and bilateral cognitive control regions, associative visual processing regions, and negative correlation with FC subcortical cognitive control, language, and V1 region
Greenwood P et al. [29]	Philips 3T	T2*	700/30	48	3	200	857	Awake, fixated on a cross	SPM 12|CONN	NR	0.008–0.09	8	MNI|WFU PickAtlas	Seed-to-voxel	CON: bilateral aPFC, lateral AI/FOp, medial AI/FOp, AI/FOp, dACC/medial SFC; FPN: bilateral DLPFC, IPL, IPS, PreCu, MCC; DAN: FEF, pIPS, aIPS, MT+, and left SMA/pre-SMA, right IFG, MFG, and AI; VAN: bilateral AI, SFG, and MTG, right SMG (TPJ), IFG/MFG, left STG, PreCu, MedFG, and sulcus callosomarginalis	No group differences	DYS shows a negative correlation between maternal education and FC between FPN and left central opercular cortex and occipital FFG
Cross AM et al. [30]	Siemens 3T	T2*	1000/30	48	3	210	360	Awake, fixated on a cross	SPM 12|CONN	FD < 0.5 mm or BSC > 3 SD	0.008–0.09	5	MNI	Seed-to-voxel	MFG, IFG opercularis, IFG triangularis, PreCG, STG posterior, AG, SPL, SMA, FFG, OP, THA	NR	Positive correlation between FC in left dorsal and anterior regions with decoding proficiency, and between left THA, and right FFG with sight word reading, RAN, and reading comprehension abilities. A negative association between reading comprehension and RSFC from SPL to SFG and MFG, in left hemisphere.
Nachshon O et al. [31]	Philips 3T	EPI	2000/38	NR	5	NR	300	Awake, fixated on a cross	SPM/CONN	NR	0.008–0.2	NR	FSL Harvard–Oxford	Seed-to-voxel and graph theory	Amygdala: BLA and CAN bilateral	DYS showed lower global efficiency values within the amygdala network, a greater FC of the bilateral amygdala with the right TP and the right MFG, and a lower FC between the left amygdala and the left FP. DYS decreases FC between the amygdala (left > right) with right FP, the left amygdala with left FP, and the right amygdala with left THA	Positive correlation between reading/EF measures and FC of left amygdala-bilateral FP, and negative correlation between emotional abilities and FC of left Amygdala-right FP
Margolis AE et al. [32]	GE 3T	EPI	2000/30	NR	3	240	292	NR	SPM 12|CONN 16b	FD > 0.5 mm or BSC > 3	0.01–0.1	8	MNI	Seed-to-voxel	CON: AI/FP, aPFC, THA bilateral and dorsal ACC FPN: DLPFC, IPL, IPS, PreCu bilaterally	DYS reduced FC of CON with left FPN and temporal-parietal regions and increased FC with cerebellum than controls. DYS showed decreases FC from FPN to DMN in controls	Both groups have a negative correlation FC of FPN (left IPL and right SFG) with Simon’s performance. DYS showed a positive correlation between FC left IPL to PCC with word-reading accuracy.
Kuhl U et al. [33]	Siemens 3T	T2*	2000/30	NR	3.99	192	75|118	NR	FSL 5.0 and AFNI 17.2.17	FD > 0.5 mm	0.01–0.1	6	MNI|Glasse atlas	ROI-to-ROI	Cortical: V1, and left MT, FG, A1, PT, BA6, BA44, and BA45/47 Subcortical areas: MGB, LGN, and iC	DYS showed lower FC in the left A1 and PT compared to controls, mainly in boys before literacy	NR
Hashimoto T et al. [34]	Philips 3T	EPI	2000/30	34	3.75	240	160	Awake eyes open	DPARSF|SPM8	FD > 0.5 mm and BSC > 1.5%	0.01–0.1	6	MNI	Seed-to-voxel	The left lFG and IPL; bilateral FG and the right IOG	No group differences.	Control: fluent reading (Hiragana) positively correlated with FC between left FG and left ventral fronto-temporal area and fewer correlations in DYS. Control > DYS showed a positive correlation between the FC of left IPL-right MFG with Kanji accuracy
Freedman L et al. [35]	Philips 3T	T2*	2000/30	NR	3	NR	300	Awake, fixated on a cross	CONN 18	NR	0.008–0.2	8	MNI|WFU PickAtlas	ROI-to-ROI	CON: bilateral aPFC and AI; FPN: bilateral DLPFC, IPL, IPS, PreCu; DAN: bilateral FEF and IPS; VAN: bilateral IFG, STG, MFG, IPL	CON and VAN FC differences between high and low fluency separation groups	NR
Davis K et al. [39]	GE 3T	EPI	2200/30	140	3.5	240	273	Minds wander, awake, looking at a cross	SPM 12 and CONN 16b	FD > 0.5 mm	0.01–0.1	6	MNI and Juelich Histological Atlas	Seed-to-voxel	Amygdala: BLA and CAN bilateral	DYS showed an increase in FC from left BLA to an MPFC compared to controls, as well as in bilateral CMA to MPFC (left SMFG and MFG)	Both groups showed FC from left BLA and bilateral CMA to mPFC positively predicted reading impairment and negatively predicted reading fluency.
Buchweitz A et al. [36]	GE 3T	EPI	2000/30	29	3.5	220	210	Awake looking at a cross	AFNI	>0.9 mm	0.01–0.1	6	Haskins Ped | MNI	Seed-to-voxel	Left FFG (VWFA), left AG, left IFG, left MTG, left STG	DYS showed lower FC between VWFA and PCC compared to controls.	NR
Horowitz-Kraus T, Wood et al. [37]	Philips 3T	EPI	2000/38	NR	5	NR	165	Awake, fixated on a cross	SPM 8|CONN 13p	FD > 2 mm	0.008–0.2	8	MNI|WFU PickAtlas	ROI-to-ROI and graph theory	CON: dACC and bilaterally aPFC, lateral AI/FOp, medial AI/FOp, and AI/FOp; FPN: bilateral DLPFC, bilateral IPL, bilateral, IPS, bilateral PreCu, mCC	No group differences in the FPN and CON	DYS had a lower reading score and higher FC FPN compared to the mood disorders group. DYS had a negative correlation between phonological awareness and FC of CON
Horowitz-Kraus T, DiCe et al. [38]	Philips 3T	EPI	2000/38	NR	5	NR	300	Awake, fixated on a cross	SPM 8|CONN	FD > 0.02 mm	0.008–0.2	8	MNI	ROI-to-ROI	Bilateral ACC/BA24	DYS showed lower FC in ACC/BA24 with left IFG/BA45	Lower negative FC between ACC and left IFG tri regions was correlated with longer fixation times during reading in DYS.
Zhou W et al. [40]	Siemens 3T	EPI	2000/30	33	3	200	240	Motionless and no thinking	DPARSF	NR	0.01–0.08	4	MNI	Seed-to-voxel	Left IPS, VWFA	DYS decreases FC between left IPS to the left MFG and VWFA to the anterior part of the left MFG compared to controls	Both groups showed a positive correlation in score of reading fluency with FC for IPS-MFG and VWFA-MFG
Horowitz-Kraus T, Toro-Serey C et al. [41]	Philips 3T	EPI	2000/38	NR	5	NR	165	Awake, fixated on a cross	SPM 8|CONN13p	FD > 2 mm	0.008–0.2	8	MNI	ROI-to-ROI	CON: bilateral aPFC, bilateral lateral AI, bilateral medial AI, bilateral AI, and dACC; FPN: mCC and bilateral DLPFC, IPL, IPS, and PreCu	No group differences in the FPN and CON	DYS showed a negative correlation between reading scores and FC of CON
Horowitz-Kraus T, Difrancesco M et al. [42]	Philips 3T	EPI	2000/38	NR	5	NR	165	Awake, fixated on a cross	SPM8 and GIFT	IC noise	NR	8	MNI	ICA/PCA	ICs— Visual: bilateral FFG, Cu, LG, ITG; Executive functions: bilateral SFG and MFG; Attention: bilateral ACC and SFG; SM: bilateral Pre and Pos CG; language: bilateral IFG, MFG, ACC, left SPL, SMG; Occipito-temporal: bilateral LG, MTG, FFG; DAN: bilateral PreCu, SFG, LG, MTG, AG; and Memory: parahippocampal gyrus	DYS showed greater FC between visual and SM IC compared with controls. DYS showed decreased FC between visual, EF, and Attention ICs compared to controls	Both groups had a negative correlation between visual and SM FC with word and nonword reading scores.
rs-fMRI in adults																	
Cheema K et al. [43]	Siemens 3T	EPI	1980/30	64	2.2	NR	182	Awake, fixated on a cross	SPM12 CONN	FD > 0.9 mm; BSC > 5 z-value;	0.01–0.1	8	MNI	Seed-to-ROI	DMN: MPFC, bilateral LP, PCC; DAN: bilateral FEF and IPS; SAL: bilateral AI and RPFC, ACC, SMG	No group differences in seeds IFG, SMG, and FFG or DMN, DAN, and SAL	The between-group difference in left SMG-left AG FC showed a positive correlation with TOWRE SWE fluency behavior, and negative correlation for DYS
Schurz M et al. [44]	Philips 1.5T	EPI	2200/45	25	5	220	110	Awake eyes open	SPM 8|CONN 13i	FD > 0.5 mm; BSC > 1.5% or	0.008–0.09	8	MNI	Seed-to-voxel	Occipito-Temporal seeds: left FFG and left ITG	DYS showed stronger FC in left FFG with right PreCu, but lower with left IFG tri, as well as between the left ITG with bilateral IFG (op, orb and tri) and stronger with left PreCu.	Positive correlation for both groups: Reading and Verbal IQ with FC of FFG with left IFG (op and tri), and ITG and STG with left IFG tri;, only reading with FC of MTG with left IFG (ob and op) and between IFG with left MTG; reading and naming was correlated with ITG and STG with left IFG op. The negative correlation was found between IFG and left IPL/AG
															Left temporo-parietal seeds: left STG and left IPL	DYS showed weaker FC in the left STG with left IFG op, and the left IPL with right hippocampus and left PreCu.	
															Left temporal and frontal seeds: left MTG and left IFG	DYS showed stronger FC in left MTG with left IFG oper, right FP and left calcarine sulcus; and lower FC in left IFG with left MTG and SMA and stronger with right rolandic operculum.	

**Note:** * FWHM measures. **Abbreviations:** Ref.: reference; fMRI: functional magnetic resonance image; NR: not reported; TR: time of repetition; TE: time of echo; ms: milliseconds; mm: millimeters; FOV: field of view; Hz: hertz; ROI: Region of interest; EPI: echo-planar imaging; DPABI: Data processing and analysis for brain imaging; FSL: FMRIB software library; MELODIC: Multivariate Exploratory Linear Optimized Decomposition into Independent Components; SPM: Statistical Parametric Mapping; Dynamic BC: Dynamic brain connectome; AFNI: Analysis of functional neuroimages; DPARSF: Data-processing Assistant for Resting-State fMRI; GIFT: Group ICA for fMRI Toolbox; FD: framewise displacement; SIC: Signal intensity changes; SD: Standard deviation; IC noise: independent components noise; FWHM: Full width at half maximum; MNI: Montreal Neurological Institute; AAL: Automated Anatomical Labeling; ICA AROMA: independent component analysis—automatic removal of motion artifacts; Dynamic FC: Dynamic Functional Connectivity; ALFF: Amplitude of low-frequency fluctuation; ReHo: Regional Homogeneity; MVPA: multi-voxel pattern analysis; ICA: independent component analysis; PCA: Principal component analysis; SOG: Superior occipital gyrus; IFG tri: inferior frontal gyrus triangularis; SMG: supramarginal gyrus; PG: Parietal gyrus; IPL: Inferior parietal lobe; SMd: Dorsal somatomotor network; SMl: Lateral somatomotor; DMN: Default mode network; PMN: Parietal memory network; VIS: Visual network; FPN: Frontoparietal network; VAN: Ventral attention network; OFN: orbitofrontal network; LMN: lateral motor network; DAN: Dorsal attention network; CON: Cingulo-opercular network; MTL: medial temporal lobe; BG: Basal ganglia; PreCG: precentral gyrus; PostCG: Postcentral gyrus; AG: angular gyrus; ITG: inferior temporal gyrus; MOG: Middle occipital gyrus; GMFA: graphemic/motor frontal area; BSC: BOLD Signal Change; BLA: Basolateral amygdaloid complex; CAN: Central amygdala nucleus; BA: Brodmann area; A1: primary auditory cortex; V1: primary visual cortex; PCC: Posterior Cingulate Cortex; ACC: Anterior Cingulate Cortex; AI: Anterior insula; aPFC: Anterior prefrontal cortex; vAI: ventral anterior insula; dACC: dorsal anterior cingulate cortex; DLPFC: dorsolateral prefrontal cortex; FEF: Frontal eye fields; IPS: intraparietal sulcus; pIPS: posterior intraparietal sulcus; aIPS: Anterior intraparietal sulcus; SMA: supplementary motor area; pre-SMA: pré-supplementary motor area; TPJ: Temporoparietal junction; SFG: Superior frontal gyrus; MedFG: medial frontal gyrus; FP: Fronto-parietal; THA: Thalamus; TP: temporal pole; MFG: middle frontal gyrus; SPL: superior parietal lobule; OP: operculum; tri: triangularis; BA17: Brodmann area 17; SAL: Salience; MT: middle temporal area; FFG: fusiform gyrus; PT: planum temporale; BA6: ventral premotor cortex; BA44: pars opercularis of the inferior frontal gyrus; BA45/47: pars triangularis/orbitalis of the inferior frontal gyrus; iC: inferior colliculus; IC: independent component; MGB: medial geniculate nucleus; LGN: lateral geniculate nucleus; IOG: inferior occipital gyrus; FOp: Frontal Operculum; VWFA: visual word form area; MTG: middle temporal gyrus; STG: superior temporal gyrus; IFG: inferior frontal gyrus; mCC: midcingulate cortex; BA24: Brodmann area 24; LG: lingula gyrus; DYS: Dyslexic; EC: effective connectivity; FC: functional connectivity; EDYS: dyslexic children who received special education; pACC: Pregenual anterior cingulate cortex; MPFC: medial prefrontal cortex; CMA: cingulate motor area; PreCu: precuneous; Cu: Cuneus; SMFG: superior medial frontal gyrus; RAN: Rapid automatized naming; EF: executive functions; RSFC: resting state functional connectivity; TOWRE: test of word reading efficiency; SWE: Sight Word Efficiency; IQ: intelligence quotient; IFG op: opercular part of the inferior frontal gyrus; Haskins Ped: Haskins pediatric atlas; CG: cingulate gyrus.; LP: Lateral posterior thalamic nucleus; SM: somatomotor network.

## Data Availability

All the data supporting the results are included in the manuscript.

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
