# Peer review of "Resting-State Functional MRI in Dyslexia: A Systematic Review"

_biomedicines, 2025, doi:10.3390/biomedicines13051210_

Round 1
Reviewer 1 Report
Comments and Suggestions for Authors
1. Line 12–15 – Objective Statement is Grammatically and Semantically Ambiguous: The phrase "addresses the systematically analyzing" is grammatically incorrect and poorly structured. This compromises clarity in stating the objective of the review.
2. Line 90–91 – Lack of PROSPERO Registration :The review was not registered in PROSPERO, which is a standard practice in systematic reviews to promote transparency, reduce risk of bias, and prevent duplication. No rationale was provided for this omission.
3. Line 122–127 – Over-Restrictive Search Criteria: The exclusion of entire subject areas and specific keywords (e.g., “Child”, “EEG”, “Alzheimer’s Disease”) may have led to the omission of relevant interdisciplinary studies. This restrictiveness may induce selection bias and limits the scope of the review. Justify the selection of MRI modalities. The omission of other modalities may not reveal the key features associated with Alzheimer's.
4. Line 185–215 – Article Selection and Inter-Rater Reliability :While the review mentions that articles were selected independently by pairs of reviewers, it does not report any metrics of inter-rater reliability (e.g., Cohen’s kappa) or describe how discrepancies were resolved quantitatively.
5. Line 237–243 – Inadequate Risk of Bias Assessment: No formal risk of bias tool (e.g., QUADAS-2, Newcastle-Ottawa Scale) was applied. The review merely states that inclusion was based on consensus, which is not a sufficient replacement for systematic quality appraisal.
6. Line 245–251 – Lack of Meta-Analytic or Thematic Synthesis : The data analysis section is limited to descriptive statistics (percentages and ranges). No thematic synthesis, subgroup analysis, or visualization techniques (e.g., forest plots, bubble plots) were used to extract deeper insights from heterogeneous findings.
7. Line 376–383 – Limited Representation of Adult Dyslexia: The review acknowledges a significant gap in adult representation (only 2 out of 25 studies), but it does not adequately discuss how this affects generalizability nor does it propose directions for future research to address this limitation.
8. Line 394–444 – Heterogeneous Neuropsychological Instruments: There is extensive variation in the neuropsychological tests used across studies without discussing how this variability impacts cross-study comparability or synthesis. Language-specific and culturally adapted tools were not critically evaluated.
9. Line 493–523 – Incomplete Reporting of Imaging Parameters: Critical acquisition parameters (e.g., head motion thresholds, scan duration, participant instructions like eyes open/closed, physiological noise correction) are inconsistently reported across studies. This hampers reproducibility and comparison.
10. Line 526 onward – Missing Justification for Analytical Choices :The rationale for choosing specific analysis types (e.g., ICA vs. ROI-to-ROI, Granger causality) is not discussed. This omission prevents the reader from evaluating the appropriateness of analytical techniques applied across studies.
11. Line 369–371 and 384–392 – Missing Educational Data and Reading Age :The majority of included studies do not report participants’ years of education or reading age. The absence of such fundamental demographic variables weakens developmental interpretations of rs-FC alterations.
12. Line 344–350 – Geographical and Linguistic Bias: There is an over-representation of studies conducted in English-speaking countries (52% in the USA), which could bias findings. This linguistic/geographic imbalance is acknowledged but not sufficiently critiqued.
13. Table 1 and Throughout – Lack of Statistical Data for Findings: While differences between dyslexic and control groups are reported, most results are not accompanied by p-values, confidence intervals, or effect sizes. This limits interpretation of the clinical and statistical significance of findings.
14. Line 355–363 – Terminology and Diagnostic Clarity: Although multiple terms are used for dyslexia (e.g., “reading disorder”, “learning impairment”), no standardized diagnostic criteria (e.g., DSM-5 or ICD-11 definitions) are discussed, which introduces inconsistency in sample characterization.
15. General – No Discussion of Publication Bias: There is no mention or analysis of publication bias (e.g., funnel plot, Egger’s test) despite being a common issue in rs-fMRI literature, particularly when studies with null findings are less likely to be published.
Comments on the Quality of English Language
Need extensive proofread. I noticed flaws in sentence formation.
Author Response
- Line 12–15 – Objective Statement is Grammatically and Semantically Ambiguous: The phrase "addresses the systematically analyzing" is grammatically incorrect and poorly structured. This compromises clarity in stating the objective of the review.
Answer: Thank you the phrase was corrected: The present review systematically analyzes the most frequently reported neuropsychological and functional connectivity
- Line 90–91 – Lack of PROSPERO Registration: The review was not registered in PROSPERO, which is a standard practice in systematic reviews to promote transparency, reduce risk of bias, and prevent duplication. No rationale was provided for this omission.
Answer: Thank you for your comment. The systematic review has been registered with PROSPERO, and this information has been added to item 2.1 of the Materials and Methods section of the manuscript, including the corresponding registration number.
- Line 122–127 – Over-Restrictive Search Criteria: The exclusion of entire subject areas and specific keywords (e.g., “Child”, “EEG”, “Alzheimer’s Disease”) may have led to the omission of relevant interdisciplinary studies. This restrictiveness may induce selection bias and limits the scope of the review. Justify the selection of MRI modalities. The omission of other modalities may not reveal the key features associated with Alzheimer's.
Answer: We appreciate the reviewer’s comment regarding the exclusion criteria. However, the scope of this review was deliberately limited to studies employing resting-state functional connectivity using MRI data, as the primary objective was to evaluate whether there is any consensus in the literature regarding alterations in resting-state functional connectivity in individuals with developmental dyslexia. We excluded studies using other neuroimaging modalities (e.g., EEG, PET, fNIRS) because each technique has distinct acquisition protocols, preprocessing pipelines, and analytical frameworks, which would make direct comparison and synthesis of results methodologically inconsistent and less meaningful within the confines of a focused systematic review.
Moreover, our aim was to explore alterations in developmental dyslexia, excluding cases where dyslexia may be secondary to other neurological conditions such as acquired brain injuries. Including such populations could introduce additional variability and confound the interpretation of connectivity alterations specifically related to developmental neurocognitive mechanisms of dyslexia.
Even within the selected rs-fMRI studies, we observed considerable heterogeneity in analytical approaches and reported findings, further justifying the importance of maintaining a focused inclusion criterion to allow for more meaningful synthesis and interpretation.
- Line 185–215 – Article Selection and Inter-Rater Reliability: While the review mentions that articles were selected independently by pairs of reviewers, it does not report any metrics of inter-rater reliability (e.g., Cohen’s kappa) or describe how discrepancies were resolved quantitatively.
Answer: Thank you for your observation. We have improved the description of the article selection process in the revised manuscript, providing clearer details on how independent screening and discrepancy resolution were conducted. Specifically, we clarified that titles and abstracts were independently reviewed in randomly assigned pairs using blinded screening, and any disagreements were resolved by a third senior reviewer to reach consensus. However, we acknowledge here that no inter-rater reliability metric (e.g., Cohen’s kappa) was calculated during the screening process.
- Line 237–243 – Inadequate Risk of Bias Assessment: No formal risk of bias tool (e.g., QUADAS-2, Newcastle-Ottawa Scale) was applied. The review merely states that inclusion was based on consensus, which is not a sufficient replacement for systematic quality appraisal.
Answer: Thank you for your observation. We would like to clarify that a formal risk of bias assessment was indeed performed using the Newcastle-Ottawa Scale (NOS). This information has now been explicitly included in the Methods section (item 2.7), where we describe the methodological quality assessment procedures applied to the included studies. Additionally, the Results section (item 3.2) presents the inter-rater agreement analysis using Cohen’s Kappa coefficient, supporting the consistency of the quality evaluations conducted independently by three reviewers. We appreciate the opportunity to clarify this point and have revised the manuscript accordingly to ensure this process is transparent to readers.
- Line 245–251 – Lack of Meta-Analytic or Thematic Synthesis : The data analysis section is limited to descriptive statistics (percentages and ranges). No thematic synthesis, subgroup analysis, or visualization techniques (e.g., forest plots, bubble plots) were used to extract deeper insights from heterogeneous findings.
Answer: Thank you for your observation. However, we would like to clarify that the objective of this work was to conduct a systematic review only, following the PRISMA (Preferred Reporting Items for Systematic Reviews and Meta-Analyses) guidelines. Given the high methodological heterogeneity among the included studies, including variability in participant characteristics, fMRI preprocessing protocols, analytical methods, and reported outcomes, conducting a proper meta-analysis was not feasible.
As a result, the data analysis section was intentionally limited to descriptive statistics (e.g., percentages and ranges), aiming to summarize methodological trends and key characteristics.
Likewise, thematic synthesis, subgroup analyses, or visualizations such as forest plots or bubble plots were not employed, as these are mostly associated with meta-analytic designs. We have revised the manuscript to clarify this distinction and better align expectations regarding the scope and analytical approach of our review.
- Line 376–383 – Limited Representation of Adult Dyslexia: The review acknowledges a significant gap in adult representation (only 2 out of 25 studies), but it does not adequately discuss how this affects generalizability nor does it propose directions for future research to address this limitation.
Answer: We appreciate the reviewer’s observation regarding the limited representation of adults in the reviewed studies. We agree that this is an important limitation and included a specific paragraph in the Discussion section to address this point. There is a clear need for more research focusing on adult populations with developmental dyslexia, particularly to understand how neuroplasticity and compensatory neural mechanisms may add to the symptomatology in adulthood after years of reading adaptation.
This limitation is not unique to the current review. In fact, it is consistent with findings from another systematic review [1] conducted by our group focused on structural MRI studies, in which only 15% of the included studies investigated adults with dyslexia. Notably, this suggests that the underrepresentation of adults is even more pronounced in functional MRI studies. Therefore, we emphasize the need for future research to address this gap, which is essential for improving the generalizability of findings and understanding the lifespan trajectory of dyslexia-related brain changes.
- Line 394–444 – Heterogeneous Neuropsychological Instruments: There is extensive variation in the neuropsychological tests used across studies without discussing how this variability impacts cross-study comparability or synthesis. Language-specific and culturally adapted tools were not critically evaluated.
Answer: We appreciate the reviewer's observation about neuropsychological test heterogeneity across studies. As we emphasized in our revised discussion, this variability reflects the essential need for cultural and linguistic adaptations to ensure ecological validity across populations. For instance, while pseudoword tasks (e.g., TOWRE) work well for opaque languages like English, character recognition tests (e.g., CRTB) prove more valid for logographic languages like Chinese. This variation doesn't represent methodological inconsistency but rather appropriate adaptations to different linguistic contexts. Also, it doesn't impose a stepback in the review because there is no intent to compare the neuropsychological tests but to describe the dyslexia´s sample main characteristics of the selected studies, considering that developmental dyslexia may be rather heterogenous impairment. These points were included in a more explicit way in the discussion.
- Line 493–523 – Incomplete Reporting of Imaging Parameters: Critical acquisition parameters (e.g., head motion thresholds, scan duration, participant instructions like eyes open/closed, physiological noise correction) are inconsistently reported across studies. This hampers reproducibility and comparison.
Answer: Thank you for your comment. We would like to clarify that all imaging acquisition parameters were carefully described and analyzed in section 3.3.1 of the Results, while details regarding image preprocessing and correction strategies are presented in section 3.3.2. To facilitate interpretation and comparison across studies, we categorized preprocessing strategies (including motion and physiological noise correction) into conservative, intermediate, or liberal approaches.
However, this explanation could be made more explicit and, therefore, the manuscript was revised to clarify why certain analytical methods (e.g., ICA, ROI-to-ROI, or Granger causality) were less frequently employed in the included studies. These choices often reflect the nature of the research question and the methodological flexibility required when studying neurodevelopmental conditions with diverse neural correlates.
- Line 526 onward – Missing Justification for Analytical Choices :The rationale for choosing specific analysis types (e.g., ICA vs. ROI-to-ROI, Granger causality) is not discussed. This omission prevents the reader from evaluating the appropriateness of analytical techniques applied across studies.
Answer: Thanks for your observation. However, we would like to clarify that the rationale for the analytical choices was indeed discussed in the manuscript. Specifically, we highlighted that most studies opted for the seed-to-voxel method, which is commonly used in exploratory analyses when there is no prior hypothesis regarding specific alterations or regions of interest. This approach allows for a broader investigation of potential functional connectivity differences across the whole brain, which is particularly useful in complex and heterogeneous conditions such as developmental dyslexia.
We acknowledge, however, that this explanation could be made more explicit. Therefore, we have revised the manuscript to strengthen this point and clarify why certain analytical methods (e.g., ICA, ROI-to-ROI, or Granger causality) were less frequently employed in the included studies. These choices often reflect the nature of the research question and the methodological flexibility required when studying neurodevelopmental conditions with diverse neural correlates.
- Line 369–371 and 384–392 – Missing Educational Data and Reading Age :The majority of included studies do not report participants’ years of education or reading age. The absence of such fundamental demographic variables weakens developmental interpretations of rs-FC alterations.
Answer: Thank you for your observation. Indeed, only five of the included studies reported participants’ years of education or reading age among children and adolescents.
However, it is important to note that the majority of studies employed neuropsychological assessments that were either age- and/or education-adjusted or normed accordingly, correlated with resting-state functional connectivity findings.
Nonetheless, we recognize that the absence of explicit reporting on education level and reading age limits a more nuanced developmental interpretation. We have added a discussion point in the revised manuscript to emphasize this limitation and the need for future studies to consistently report these key demographic and cognitive variables.
- Line 344–350 – Geographical and Linguistic Bias: There is an over-representation of studies conducted in English-speaking countries (52% in the USA), which could bias findings. This linguistic/geographic imbalance is acknowledged but not sufficiently critiqued.
Answer: We thank the reviewer for raising this important point. We agree that the geographical and linguistic concentration of studies, particularly the over-representation of research conducted in English-speaking countries (52% from the USA), introduces a potential bias that deserves further critical attention. To address this, we have expanded the discussion to include an analysis of the geopolitical and linguistic landscape of current dyslexia research, which reflects not only access to research infrastructure but also broader publication trends.
A recent bibliometric analysis identified 9,166 publications on dyslexia from 2000 to 2021 indexed in the SSCI and SCI-E. The United States, the United Kingdom, and Germany emerged as the top three most productive countries, while China, Israel, and Japan were leaders in Asia. The University of Oxford stood out as the institution with the highest output and h-index, and the journal Dyslexia was the most prolific in the field, followed by journals in Psychology. These findings reflect the dominance of Western institutions and English-language publications, which may skew the global understanding of dyslexia, particularly when it comes to generalizing neurobiological patterns. [2]
Another relevant aspect highlighted in our review is the predominance of studies conducted in opaque languages, which represented 64% of the selected studies, mainly in English and Chinese. These languages exhibit less consistent phoneme-grapheme correspondence, which complicates the decoding process and may accentuate the clinical manifestations of dyslexia. In contrast, transparent languages such as Italian, Spanish, and Portuguese tend to exhibit distinct reading profiles in dyslexia, where difficulties are more often associated with reading speed than accuracy. This linguistic variability reflects different phenotypic expressions of dyslexia, and the current overrepresentation of opaque language contexts may limit the representativeness and generalizability of the findings.
We have now incorporated these points into the revised manuscript, including a dedicated paragraph in the Discussion section to critically contextualize this linguistic and geographic imbalance and its implications for the interpretation of resting-state fMRI findings in dyslexia.
- Table 1 and Throughout – Lack of Statistical Data for Findings: While differences between dyslexic and control groups are reported, most results are not accompanied by p-values, confidence intervals, or effect sizes. This limits interpretation of the clinical and statistical significance of findings.
Answer: Thank you for your comments. However, we would like to clarify that the objective of this article was to conduct a detailed systematic review following the PRISMA guidelines, rather than a meta-analysis. A meta-analytic approach was not feasible due to the high methodological heterogeneity among the included studies, particularly regarding sample characteristics, imaging acquisition protocols, analytical methods, and outcome measures.
Nevertheless, descriptive statistical data were provided to characterize the included studies, as shown in Table 1. The analysis and interpretation of these methodological features are thoroughly described in the results section. We have clarified this point in the manuscript to better align expectations regarding the scope and analytical strategy of the review.
- Line 355–363 – Terminology and Diagnostic Clarity: Although multiple terms are used for dyslexia (e.g., “reading disorder”, “learning impairment”), no standardized diagnostic criteria (e.g., DSM-5 or ICD-11 definitions) are discussed, which introduces inconsistency in sample characterization.
Answer: Thank you for your observation. We agree that the lack of standardized diagnostic terminology can lead to inconsistency in sample characterization across studies. In response, we will include in the introduction a paragraph clarifying the diagnostic criteria for developmental dyslexia based on established classifications, such as the DSM-5 and ICD-11. This addition will help contextualize the terminology used across the reviewed studies and emphasize the importance of consistent diagnostic frameworks for interpreting neuroimaging findings in dyslexia research.
- General – No Discussion of Publication Bias: There is no mention or analysis of publication bias (e.g., funnel plot, Egger’s test) despite being a common issue in rs-fMRI literature, particularly when studies with null findings are less likely to be published.
Answer: Thank you for your comments. However, the primary aim of this work was to conduct a systematic review, following the PRISMA guidelines, and not a meta-analysis, which would typically require a quantitative synthesis of effect sizes and permit formal assessments of publication bias (e.g., funnel plots, Egger’s test).
As a result, we did not include formal tests for publication bias. Instead, we focused on evaluating the methodological quality and variability of the included studies, which we believe provides relevant insights into the current state of the literature. We have clarified this rationale in the revised version of the manuscript to better align reader expectations regarding the scope of the review.
References
- Martins, B.; Baba, M.Y.; Dimateo, E.M.; Costa, L.F.; Camara, A.S.; Lukasova, K.; Nucci, M.P. Investigating Dyslexia through Diffusion Tensor Imaging across Ages: A Systematic Review. Brain Sci 2024, 14, doi:10.3390/brainsci14040349.
- Wu, Y.; Cheng, Y.; Yang, X.; Yu, W.; Wan, Y. Dyslexia: A Bibliometric and Visualization Analysis. Frontiers in public health 2022, 10, 915053, doi:10.3389/fpubh.2022.915053.

Reviewer 2 Report
Comments and Suggestions for Authors
An interesting systematic review on fMRI resting state connectivity in those with dyslexia. The authors sought to find scientifically sound articles on this area of study and hoped to find greater numbers of articles relating to adults with Dx of dyslexia. Of the 25 articles found, fitting criteria, only 2 were adult based studies.
The findings relating to areas of reduced connectivity in the left hemisphere and the possibility of increased, compensatory connectivity in the right hemisphere is interesting and intellectually appealing but the evidence submitted is not compelling enough to be definitive.
Also the discussion about the Dx of dyslexia and its presentation in different culture with transparent vs. opaque languages is informative.
The conclusions rendered by the authors do reflect the findings of the systemic review and do point to the need for more targeted studies in adults with dyslexia as well as the heterogeniety in the methodology of the studies reported does point to the need for further refinement in Dx. and reporting of this diagnosis and the need for sequential/developmental studies to examine developmental and neuroplasticity changes in those with dyslexia vs. control norms.
Author Response
An interesting systematic review on fMRI resting state connectivity in those with dyslexia. The authors sought to find scientifically sound articles on this area of study and hoped to find greater numbers of articles relating to adults with Dx of dyslexia. Of the 25 articles found, fitting criteria, only 2 were adult based studies.
- The findings relating to areas of reduced connectivity in the left hemisphere and the possibility of increased, compensatory connectivity in the right hemisphere is interesting and intellectually appealing but the evidence submitted is not compelling enough to be definitive.
Answer: Thank you for this comment we agree that our review could not provide compelling evidence on the compensatory role of right-hemisphere activation and/or conectivity. Although there are studies that show an increased activation and connectivity in right-hemisphere, particularly the right inferior frontal gyrus and right occipito-temporal areas (Pugh et al., 2000; Richlan, 2012; Krafnick et al., 2011) these are out of the scope of our revision. We have clarified these points in the revised manuscript and included additional references to better frame the evidence as suggestive but not conclusive.
- Also the discussion about the Dx of dyslexia and its presentation in different culture with transparent vs. opaque languages is informative.
Answer: We appreciate your comment regarding the interaction between dyslexia, culture, and language. In response, we have expanded the discussion in the revised manuscript to provide a deeper exploration of how transparent versus opaque languages may influence patterns of dyslexia. This expanded section is now supported by recent studies, neurocognitive models, and neurofunctional findings.
- The conclusions rendered by the authors do reflect the findings of the systemic review and do point to the need for more targeted studies in adults with dyslexia as well as the heterogeniety in the methodology of the studies reported does point to the need for further refinement in Dx. and reporting of this diagnosis and the need for sequential/developmental studies to examine developmental and neuroplasticity changes in those with dyslexia vs. control norms.
Answer: Thank you. We hope that our review will contribute to greater consistency and comparability across studies on dyslexia.

Round 2
Reviewer 1 Report
Comments and Suggestions for Authors
The current version of this manuscript has been significantly improved. I thank the authors for their effort.